# Historical and projected future runoff over the Mekong River Basin

Chao Wang[1], Stephen Leisz[2, 3], Li Li[4], Xiaoying Shi[5], Jiafu Mao[5], Yi Zheng[1], and Anping Chen[6]

[1]School of Environmental Science and Engineering, Southern University of Science and Technology, Shenzhen 518055, China
[2]Department of Anthropology and Geography, Colorado State University, Fort Collins, CO 80523, USA
[3]VinUniversity, Ocean Park, Hanoi, Vietnam
[4]Department of Civil and Environmental Engineering, The Pennsylvania State University, University Park, PA 16802, USA
[5]Environmental Sciences Division and Climate Change Science Institute, Oak Ridge National Laboratory, Oak Ridge, TN 37831, USA
[6]Department of Biology and Graduate Degree Program in Ecology, Colorado State University, Fort Collins, CO 80523, USA

**Correspondence:** Anping Chen (anping.chen@colostate.edu)

**Abstract.** The Mekong River (MR) crosses the borders and connects six countries including China, Myanmar, Laos, Thailand, Cambodia, and Vietnam. It provides critical water resources and supports natural and agricultural ecosystems, socio-economic development, and livelihoods of the people living in this region. Understanding changes in runoff of this important international river under projected climate change is critical for water resource management and climate change adaptation planning. However, research on long-term runoff dynamics for the MR and the underlying drivers of runoff variability remains scarce. Here, we analyse historical runoff variations from 1971 to 2020 based on runoff gauge data collected from eight hydrological stations along the MR. With these runoff data, we then evaluate the runoff simulation performance of five global hydrological models (GHMs) forced by four global climate models (GCMs) under the ISIMIP project. Furthermore, based on the best simulation combination, we quantify the impact of future climate change on river runoff changes in the MR. The result shows that the annual runoff in the MR has not changed significantly in the past five decades, while the establishment of dams and reservoirs in the basin visibly affected the annual runoff distribution. The ensemble-averaged result of WaterGap2 (i.e., GHM) forced by four GCMs has the best runoff simulation performance. Under representative concentration pathways (RCPs, i.e., RCP2.6, RCP6.0 and RCP8.5), runoff of the MR is projected to increase significantly ($p<0.05$), e.g., $3.81\pm3.47\ \mathrm{m^3\ s^{-1}\ a^{-1}}$ ($9\pm8\%$ increase in 100 years) at the upper reach under RCP2.6 and $16.36\pm12.44\ \mathrm{m^3\ s^{-1}\ a^{-1}}$ ($13\pm10\%$ increase in 100 years) at the lower reach under RCP6.0. In particular, under the RCP6.0 scenario, the increase in annual runoff is most pronounced in the middle and lower reaches due to increased precipitation and snowmelt. Under the RCP8.5 scenario, the runoff distribution in different seasons varies obviously, increasing the risk of flooding in the wet season and drought in the dry season.

## 1 Introduction

Earth has been experiencing unprecedented climate change since the 1950s (IPCC, 2021). Changes in the climate system are expected to lead to regionally divergent alterations in the hydrological cycle (Giuntoli et al., 2015; Prudhomme et al., 2014). In particular, with the $CO_2$-induced increase in radiative forcing, global runoff is expected to increase (Milly et al.,

2005; Yang et al., 2017; IPCC, 2021). Yet the change is also highly heterogenous across different regions (Arnell et al., 2011; Yang et al., 2017). For example, while large runoff increases are expected in moist tropics and high latitudes, dry tropical regions are likely to experience a decrease in runoff (Hagemann et al., 2013; Field and Barros, 2014; Schewe et al., 2014).

Moreover, obvious uncertainties also exist for projected changes in regional and global runoff. Coupled state-of-the-art global climate models (GCMs) and global hydrological models (GHMs) are increasingly used for assessments of changes in the hydrological cycle (Li et al., 2017; Krysanova et al., 2018; Wang et al., 2021). Different GCMs use distinct representations of the climate system, leading to "climate model structural uncertainty" (Gosling and Arnell, 2011). Furthermore, differences in GHM structures could also result in large uncertainties in modelled runoffs. In particular, GHMs are modelled on a global

scale, and most GHMs are not calibrated. It is common that the performance of GHMs tends to vary with regional location and catchment size (Krysanova et al., 2018). Because simulated river runoffs can guide policy decisions regarding regional water resource management and climate change adaption (Arnell and Gosling, 2016), assessing model performance and reducing uncertainties in modelling results are especially desired at regional scales (Krysanova et al., 2018).

The Mekong River (MR) is an important international river running from the Tibetan Plateau through China and the coun-
35 tries of Mainland Southeast Asia (i.e., Myanmar, Laos, Thailand, Cambodia, and Vietnam) before emptying through southern Vietnam into the South China Sea (Liu et al., 2020). The upper reaches of the MR, located in China, is called the Lancang River (Wang et al., 2021) and the lower reaches are known as they pass through each country as follows; in Laos it is Mènam Khong, in Thailand it is Mae Nam Khong, in Cambodia it is Mékôngk, and in Vietnam it is Song Tien Giang (https://www.britannica.com/place/Mekong-River, accessed February 23, 2023). The production and life of the residents along
the river are directly affected by the changes in the water volume of the MR. Seasonally the water from the Mekong flood pulse, when the river backs up and floods the Ton Le Sap in Cambodia, is responsible for local fish raising that provides up to 70% of Cambodians' animal protein intake and also allows for the growing of floating rice which feeds the communities in central Cambodia (Eyler, 2019). Furthermore, the fish that originate in the Ton Le Sap leave the lake as the water level decreases and restock a large part of the lower Mekong reach (Eyler, 2019) providing an important economic source of income
for the population of this area. The Mekong also provides the freshwater necessary for growing rice in the delta in Vietnam, which is considered the "rice bowl" of Vietnam (Tran et al., 2018).

The main stream of the MR extends by 4,800 $\mathrm{km}$, with a drainage area of about 795,000 $\mathrm{km^2}$ (Adamson et al., 2009). The average annual runoff at the outlet is 14,500 $\mathrm{m^3\ s^{-1}}$, making it the tenth largest river of the world in term of water discharge (Cochrane et al., 2014). However, the performance of GHMs for the Mekong River Basin (MRB) has rarely been reported
(Chen et al., 2021), which could impede improved predictions of future runoffs. Importantly, while a number of models have been used to simulate the runoff of the MR (Johnston and Kummu, 2012; Kingston et al., 2011; Li et al., 2017; Yun et al., 2020; Wang et al., 2021), these studies focus on the simulation and analysis of the MR runoff under different climate models but by a single hydrological model without comparing performances of different hydrological models. On the other hand, Chen et al. (2021) assesses the applicability of ten hydrological models in the MR using one set of meteorological forcing
data from the Global Soil Wetness Project 3 (GSWP3) under the ISIMIP project. Their study shows that the calibrated GHMs have the best performances during the historical scenarios period. However, these studies do not systematically analyse the

runoff simulation results of long-term historical periods (including the historical period of historical scenarios and the real-time period of representative concentration pathways (RCP) scenarios, i.e., from the start simulation year of the RCPs to 2020, for which observed runoff data are available) under different GCM-GHM combinations. In the context of high uncertainty
in runoff projections under the RCPs, the use of real-world observations to evaluate future projections during the real-time period can increase the reliability of the simulation for more distant future periods. Such an analysis is meaningful and urgent to potentially assess and reduce the uncertainty/bias of runoff simulations introduced by both GCMs and GHMs to achieve more reliable future projections (Kingston et al., 2011; Hoang et al., 2016). As one of the longest rivers in the world and one that is the major water source for 65 million people in the five countries of the Lower Mekong Region, comprehensive model
evaluation for the MR runoff is critical in order to understand the limitations and strengths of, and further improve, the global hydrology models for wide application and for better policy decisions for the regions.

The goal of this study is to understand the temporal and spatial variation characteristics of the MR runoff, with a focus on the future runoff changes under different RCP scenarios. To this end, we perform the following analyses: (1) We first perform a trend analysis and significance test on the historical observed runoff during the period $1971-2020$ at the eight gauging stations
located along the mainstream (Fig. 1 ). (2) We then evaluate the runoff simulations of different GCM-GHM combinations in ISIMIP over historical scenario periods and real-time periods of RCPs against observed runoff from the above gauging stations, and identify the best GCM-GHM combination for predicting future runoff changes. (3) Finally, we comprehensively analyse the future runoff pattern changes (including annual runoff and seasonal runoff) in the upper, middle and lower reaches of the MRB under future RCP scenarios based on the best GCM-GHM combination.

## 2    Materials and Methods

### 2.1    Study area and hydrological stations

Located between 9° and 35° north and 94° and 110° east (Fig. 1), the MR drains water from a rather narrow basin area. The river is commonly divided into upper and lower parts at the China-Laos boundary. The Lower Mekong River is about 2,668 km in length (about 55.6% of the total length), but the Lower Mekong River Basin (LMRB) accounts for nearly 80% of the
total drainage of the MRB. The Mekong River Commission (MRC) manages a data base of dozens of gauging stations that monitor the runoff of the mainstream and tributaries of the LMRB. For the availability and completeness of the data time series, we select eight hydrological stations from the upper, middle, and lower reaches of the Lower Mekong River, including Chiang Saen (N1), Chiang Khan (N2), Nong Khai (N3), Nakhon Phanom (N4), Mukdahan (N5), Khong Chiam (N6), Pakse (N7), and Stung Treng (N8) (Fig. 1). The latitude and longitude locations and annual runoff of the hydrological stations are
provided in Table 1. Monthly observed runoff data from MRC (https://portal.mrcmekong.org/) ranging from 1971 to 2020 serve as validation data for the GHMs.

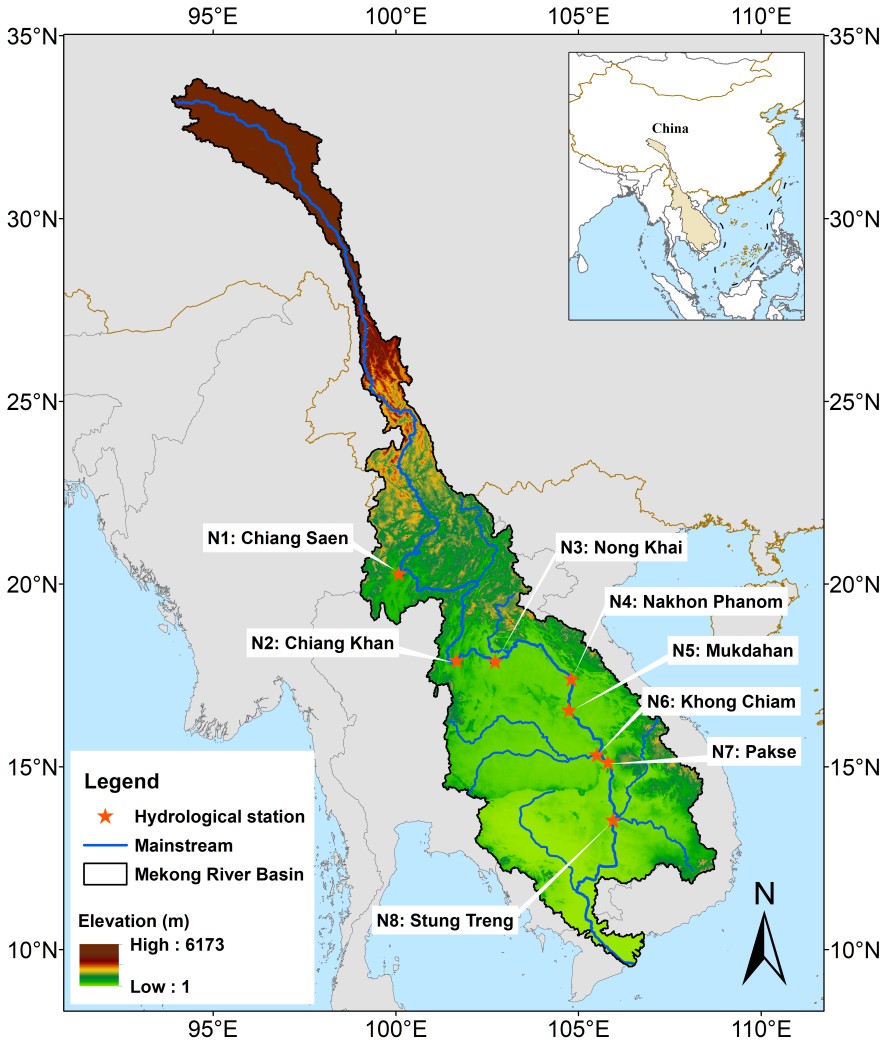

**Figure 1.** The Mekong River Basin and the locations of the eight hydrological stations used in modelling comparisons. Note all the stations are in the Lower Mekong River Basin. See Table 1 for detailed station information.

**Table 1.** Basic statistical information of the eight hydrological stations. Note Upstream, Midstream, and Downstream are referred to the stations' locations in the Lower Mekong River.

| Number | Station | Country | Location | Latitude (°) | Longitude (°) | Annual runoff ($m^3 \, s^{-1}$) |
|--------|---------|---------|----------|--------------|---------------|--------------|
| N1 | Chiang Saen | Thailand | Upstream | 20.27 | 100.08 | 2582 |
| N2 | Chiang Khan | Thailand | Upstream | 17.90 | 101.67 | 4309 |
| N3 | Nong Khai | Thailand | Upstream | 17.88 | 102.72 | 4405 |
| N4 | Nakhon Phanom | Thailand | Midstream | 17.40 | 104.80 | 7514 |
| N5 | Mukdahan | Thailand | Midstream | 16.54 | 104.74 | 7870 |
| N6 | Khong Chiam | Thailand | Downstream | 15.32 | 105.50 | 9013 |
| N7 | Pakse | Laos | Downstream | 15.12 | 105.80 | 9819 |
| N8 | Stung Treng | Cambodia | Downstream | 13.53 | 105.95 | 12677 |

## 2.2 Trend test method

Both linear regression models and the Mann-Kendall test (MK test) are commonly used to test the linear trend of annual climatic variables and the significance of variation of trends. The MK test is a noparametric method for detecting trends in time series with minimal assumptions (Lv et al., 2019) and has been widely applied to test trends in hydrological and meteorological series around the world. Compared to parametric tests (e.g., regression coefficient test), non-parametric tests (e.g., the MK test) have no requirements of homoscedasticity or prior assumptions on the distribution of the data sample (Bihrat and Bayazit, 2003) and are less sensitive to outliers (Hamed, 2007). As the MK test statistic is determined by the ranks and sequences of time series rather than the original values, it is robust when dealing with non-normally distributed data, which are commonly encountered in hydrometeorological time series (Wang et al., 2020). The MK test statistic index $U$ (referred to as MK value) follows the standard normal distribution. A positive or negative $U$ value indicates whether the trend is increasing or decreasing. The null hypothesis in this test is that there is no significant trend in the time series at the significance level of p. If $|U| > U_{\frac{p}{2}}$, where $U_{\frac{p}{2}}$ is the critical value of the standard normal distribution with a probability greater than $\frac{p}{2}$, then the null hypothesis is rejected and the trend is significant (Guan et al., 2021). This study adopts the significance level of 0.05, which means that there is a significant trend of change when the p-value is less than 0.05.

## 2.3 Climate projections and hydrological models

The Inter-Sectoral Impact Model Intercomparison Project (ISIMIP) is a community-driven modelling effort and offers a framework for comparing climate impact projections in different sectors and at different scales (Warszawski et al., 2014). Specifically, the ISIMIP2b scenarios are designed to elicit the contribution of climate change to impacts arising from low-emissions climate-change scenarios (Frieler et al., 2017). The global climate models (GCMs) selected for this study are derived from the ISIMIP2b protocol, which provides four GCMs from CMIP5 and three emission scenarios (i.e., RCP2.6, RCP6.0 and

**Table 2.** Basic information of the GHMs in the ISIMIP2b Global Water program. The runoff simulation results of the GHMs forced by different GCMs are all derived from the ISIMIP protocol (https://data.isimip.org/search/product/).

| Impact model | Meteorological forcings [1] | Evapotranspiration scheme | Snow scheme | Routing scheme |
|---|---|---|---|---|
| CLM4.5 | tas, pr, sfcWind, rlds, rsds, huss | Absent | Snow model | MOSART model |
| H08 | tas, rlds, rsds, prsn, ps, pr | Bulk formula | Energy balance | Based on DDM30 |
| LPJml | tas, rsds, pr | Priestley Taylor | Degree-day method | Linear reservoir model based on DDM30 |
| MATSIRO | ta, huss, prsn, ps, pr, tasmax, tasmin,tas, rlds, rsds, sfcWind | Constant stomatal resistance based on Farquhar-type model | Surface energy balance method | TRIP model based on DDM30 |
| WaterGAP2 | tas, rlds, rsds, pr | Priestley Taylor | Degree-day method | Linear reservoir model based on DDM30 |

[1] Notes: tas: air temperature, huss: Near-surface specific humidity, sfcWind: Near-surface wind speed, rlds: long wave downwelling radiation, rsds: short wave downwelling radiation, pr: total precipitation, tas: daily mean temperature, prsn: snowfall, ps: surface air pressure, tasmax: daily maximun temperature, tasmin: daily minimum temperature.

RCP8.5). In particular, the four GCMs are the Model for Interdisciplinary Research on Climate 5 (MIROC5), the Hadley Global Environment Model 2-Earth System (HadGEM2-ES), the Geophysical Fluid Dynamics Laboratory's Earth System Model 2 M (GFDL-ESM2M) and the Institute Pierre Simon Laplace Climate Model 5A Low Resolution (IPSL-CM5A-LR). These climate models are used because they provide detailed daily climate data, fine spatial scale and they have shown good performance in reproducing historical precipitation conditions in the MRB (Ul Hasson et al., 2016; Wang et al., 2021).

This study has selected five global hydrological models (GHMs) to evaluate the runoff simulations in the MRB, and they are the Water Global Assessment and Prognosis version 2 (WaterGAP2) (Alcamo et al., 2003; Müller Schmied et al., 2016), the Lund-Potsdam-Jena managed Land (LPJmL) (Sitch et al., 2003), the H08 (Hanasaki et al., 2018), the Community Land Model version 4.5 (CLM4.5) (Leng et al., 2015) and the Minimal Advanced Treatments of Surface Interaction and Runoff (MATSIRO) (Takata et al., 2003). The ensemble-averaged results of the GHMs model are also added to the validation (Chen et al., 2021). Table 2 shows the daily meteorological forcing variables and the main physical process modules of the above five GHMs. All GHMs operate under the meteorological forcing of the four GCMs, and the ensemble-averaged results of the GCMs are also evaluated due to the variability of the GCMs and the uncertainty of climate change. The standard deviation of the outputs of the GHM driven by four GCMs is used to quantify the uncertainty from the GCMs. The runoff simulation results of five GHMs forced by four GCMs are all derived from the experimental data of the global water sector in ISIMIP2b.

## 2.4 Model validation and performance indices

Simulated monthly runoffs from different combinations of GCM-GHM are used to validate a monthly time series for each gauge station (Table 1). For these simulated data, by combining the runoff data of both the historical (1850−2005) and the future RCPs (2006−2099) scenarios, we obtain the series corresponding to the same period (1971−2020) as the observed runoff data. We choose to verify the historical phase of the historical simulation (1971−2005) and the historical phase of the RCPs simulation (2006−2020) separately. Runoff validation during RCPs period is particularly important given the uncertainties of climate change under the future projection, and good model performance would greatly increase our confidence in future runoff simulations (Krysanova et al., 2018). For model performance metrics, we select three matrices of quantitative statistics, including: Pearson's correlation coefficient squared ($R^2$) Eq. (1), Nash-Sutcliffe efficiency (NSE) Eq. (2), and the percentage bias (Pbias) Eq. (3):

$$R^2 = \frac{[\sum_{t=1}^{T}(Q_{obs}^t - \bar{Q_{obs}}) \times (Q_{sim}^t - \bar{Q_{sim}})]^2}{\sum_{t=1}^{T}(Q_{obs}^t - \bar{Q_{obs}})^2 \times \sum_{t=1}^{T}(Q_{sim}^t - \bar{Q_{sim}})^2} \tag{1}$$

$$NSE = 1 - \frac{\sum_{t=1}^{T}(Q_{obs}^t - Q_{sim}^t)^2}{\sum_{t=1}^{T}(Q_{obs}^t - \bar{Q_{obs}})^2} \tag{2}$$

$$Pbias(\%) = 100 \times \frac{\sum_{t=1}^{T} Q_{sim}^t - \sum_{t=1}^{T} Q_{obs}^t}{\sum_{t=1}^{T} Q_{obs}^t} \tag{3}$$

where $Q_{obs}^t$ is the runoff observation value at time t, $Q_{sim}^t$ is the runoff simulation value at time t. And T is the total timesteps.

The values of $R^2$ varies between 0 and 1 and reflect the quality of the model for simulating the flow time trend. The closer $R^2$ is to 1, the stronger the simulation ability of the model. NSE is a common evaluation index in the field of hydrology. Its value range is (-∞, 1]. An NSE value close to 1 indicates good model performance while that close to 0 means credible model performance but with large errors. Negative NSE values mean that the model is not credible. Pbias shows the average trend of overestimation or underestimation of the model results compared with the observed data. The closer to 0, the smaller the model deviation and the more credible the results.

## 3 Results

### 3.1 Observation-based historical runoff changes

Fig. 2 show the annual runoff trend and significance test results of each station from 1971 to 2020. From the interannual variation process and trend line, the upstream stations (N1−N3) and downstream stations (N6−N8) have a decreasing trend, while the midstream stations (N4−N5) have an increasing trend. However, these changes are not significant (p>0.05) in the long-term trend except for N3, an result consistent with the findings by Li et al. (2017). Based on the 5-year moving average,

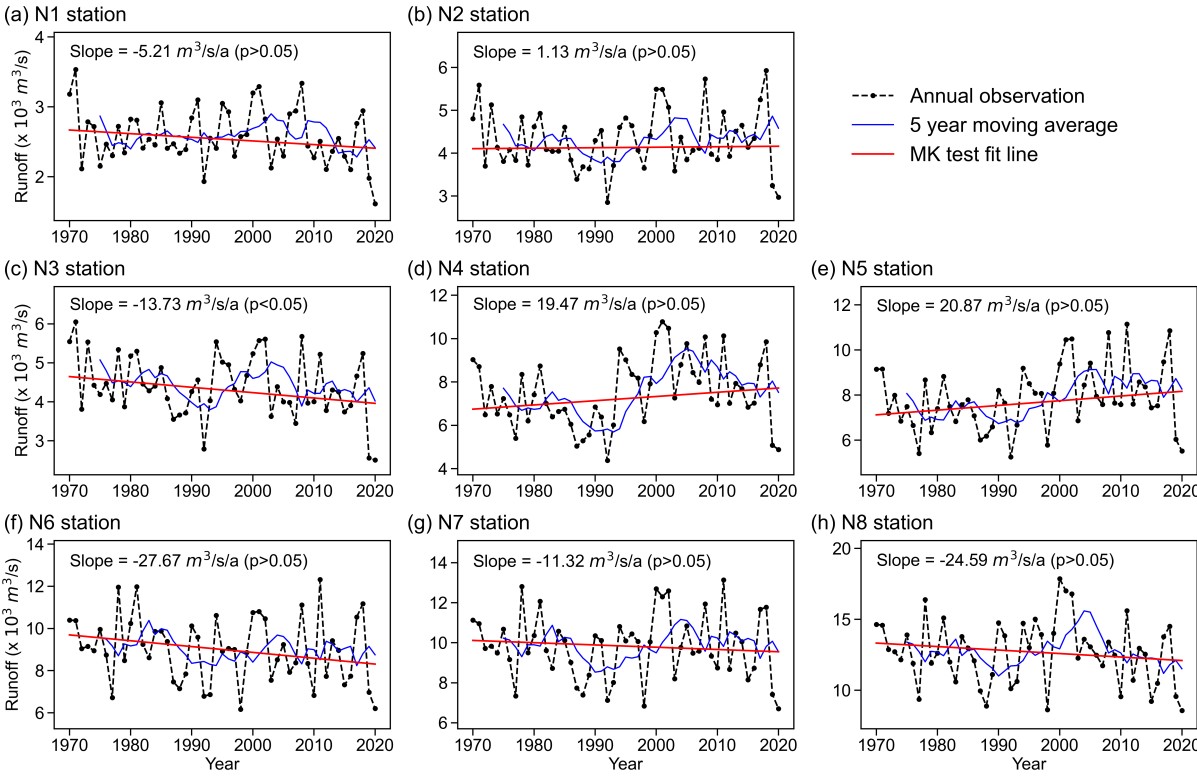

**Figure 2.** The results of the MK trend test in historical (1971-2020) runoff over the eight hydrological stations. Eight hydrological stations are numbered N1-N8, with their locations presented in the map of Fig. 1.

the runoff from the middle and lower reaches has a steep decline and then a slow rise in the 1990s, which is closely related to the construction of dams and reservoirs during this period (Lu and Siew, 2006). Some studies have shown that in the early stage of the construction of a reservoir, the impoundment of the reservoir will have an impact on the annual runoff (Lu et al., 2014). However, during the operation scheduling period after the completion of the reservoir impoundment, its impact on the annual runoff is relatively small, although the impact on the annual distribution of runoff is relatively large (Lu et al., 2014).

### 3.2 Verifying ISIMIP historical and future projections

#### 3.2.1 Historical scenario (1971–2005) simulation performance

During the validation of the historical scenarios of ISIMIP, the simulated runoff under different GCM-GHM combinations with the measured discharges at the hydrological stations are compared and it is found that most of the combinations performed well (Fig. 3). This indicates that even if GHMs instead of regional hydrological models are chosen, GHMs still have satisfactory performance in runoff simulations. As far as the differences among GCMs are concerned, except for GFDL-ESM2M, the simulation results of GHMs driven by all other climate models are generally good. Krysanova et al. (2018) suggest us-

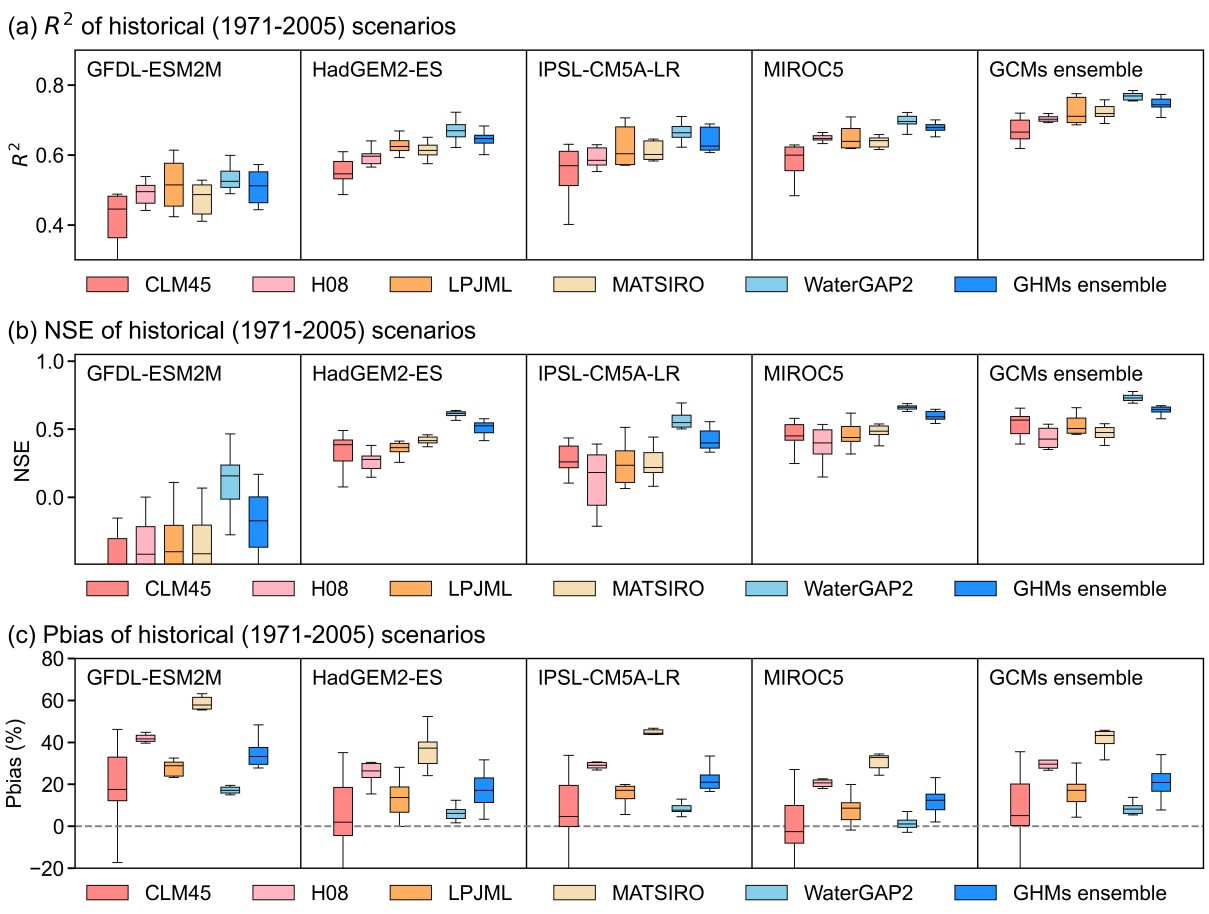

**Figure 3.** Performance of all combinations of GCMs and GHMs during historical (1971-2005) periods. The three rows correspond to three performance matrices ($R^2$, NSE and Pbias) of all GCM-GHM combinations at the eight hydrological stations. In each row, each panel is for a different GCM, as annotated. Then in each panel, the different colors are for each GHM, as marked under each row.

ing ensemble-averaged results from multiple GCMs to reduce climate model uncertainty. The results here show that GCMs ensemble-averaged simulations have an overall higher accuracy than that of individual GHMs results, and the model confidence also increases. In addition, all of the GHMs show $R^2$ at least 0.6 under GCMs ensemble-averaged, demonstrating a model performance that is reasonably good. Among these GHMs, WaterGap2 has the highest $R^2$ and NSE and a lower Pbias than others under the same GCM forcing. This is in line with the findings of Chen et al. (2021) that points out that the WaterGAP2 model is more suitable for the runoff simulation in the MRB than other models. In particular, the combination of GCMs ensemble averaging and WaterGAP2 performs the best for runoff simulations. The evaluation indicators are: $R^2 = 0.78$ $(0.72-0.82)$, NSE $= 0.68$ $(0.63-0.81)$, Pbias $= 5.5\%$ $(4.2\%-10\%)$. Generally speaking, a distributed regional hydrological model specially developed for a region will be more suitable for the simulation and evaluation of water resources in the region. However, the simulated performance under this combination is comparable to the evaluation index reported by Wang et al. (2021) which use

 the distributed hydrological model (SWAT). Good temporal dynamic capture (average $R^2$ of 0.78, average NSE of 0.68) and extremely low total runoff volume bias (average Pbias of 5.5 %) indicate that the combination of GCMs ensembled average and WaterGAP2 is likely to produce the most reliable runoff simulations for this region.

### 3.2.2  Different RCPs (2006-2020) simulation performance

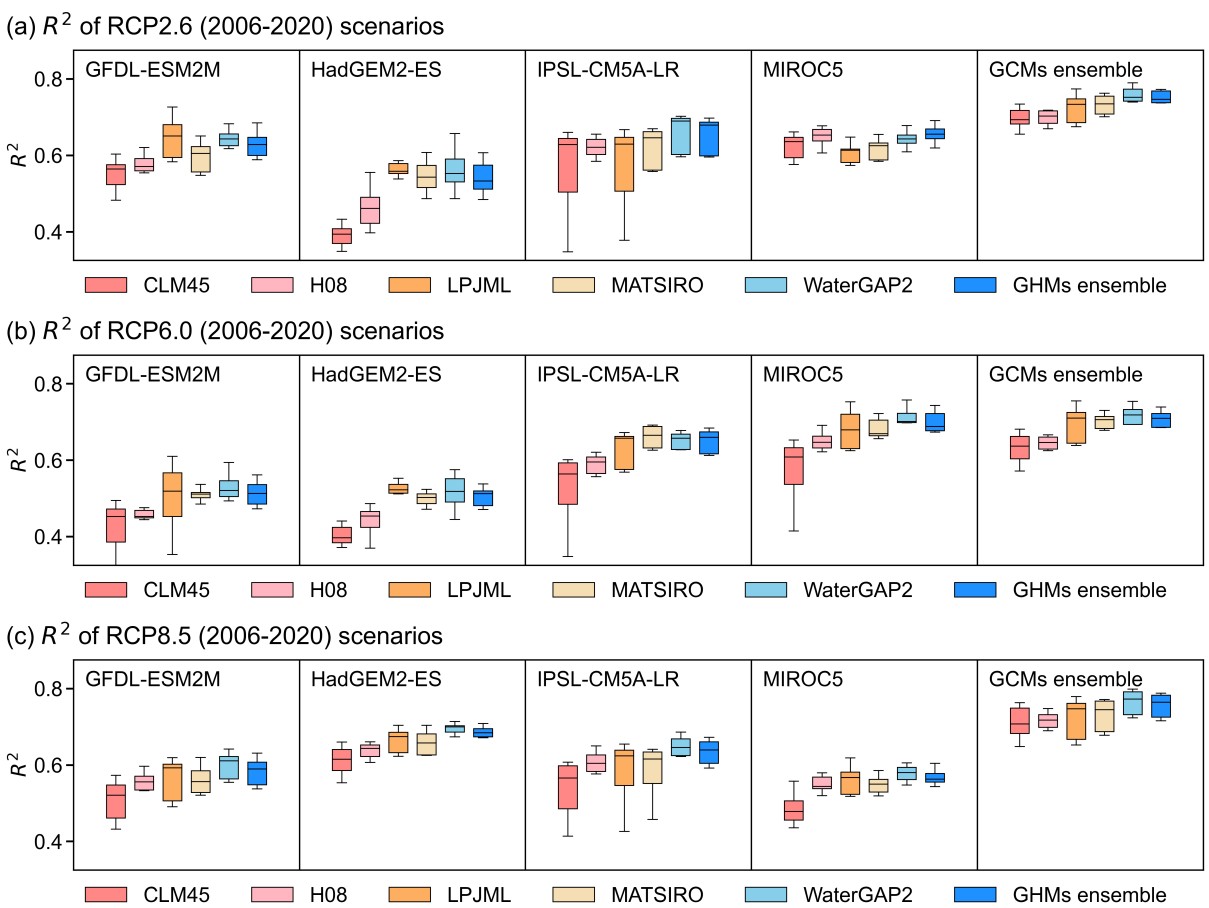

**Figure 4.** The $R^2$ performance matric of ISIMIP historical $(2006-2020)$ modelling under different RCP scenarios. The three rows correspond to three RCP scenarios (RCP2.6, RCP6.0 and RCP8.5) of all GCM-GHM combinations at the eight hydrological stations. In each row, each panel is for a different GCM, as annotated. Then in each panel, the different colours are for each GHM, as marked under each row.

ISIMIP2b projections are published before 2006 so its future projections include the period $2006-2020$, a period that now  has real-time/world observations to test against the projections. The simulation performance of these GCM-GHM combinations during this time is thus further evaluated. Similar to the historical scenario verification process, the results under RCPs scenarios with different GCM-GHM combinations are verified and compared (Fig. 4). The results of this work show that under

three mission pathways (RCP2.6, RCP6.0 and RCP8.5), the runoff simulation performance of each GCM-GHM combination is consistent with the runoff performance under the historical scenario (1971−2005). The combination of GCMs ensemble and WaterGAP2 again performs the best, with $R^2$ at least 0.70 under three mission pathways. This model combination can accurately simulate the runoff process in the real-time period under the future scenarios, which increases the reliability of the simulation for the further future period.

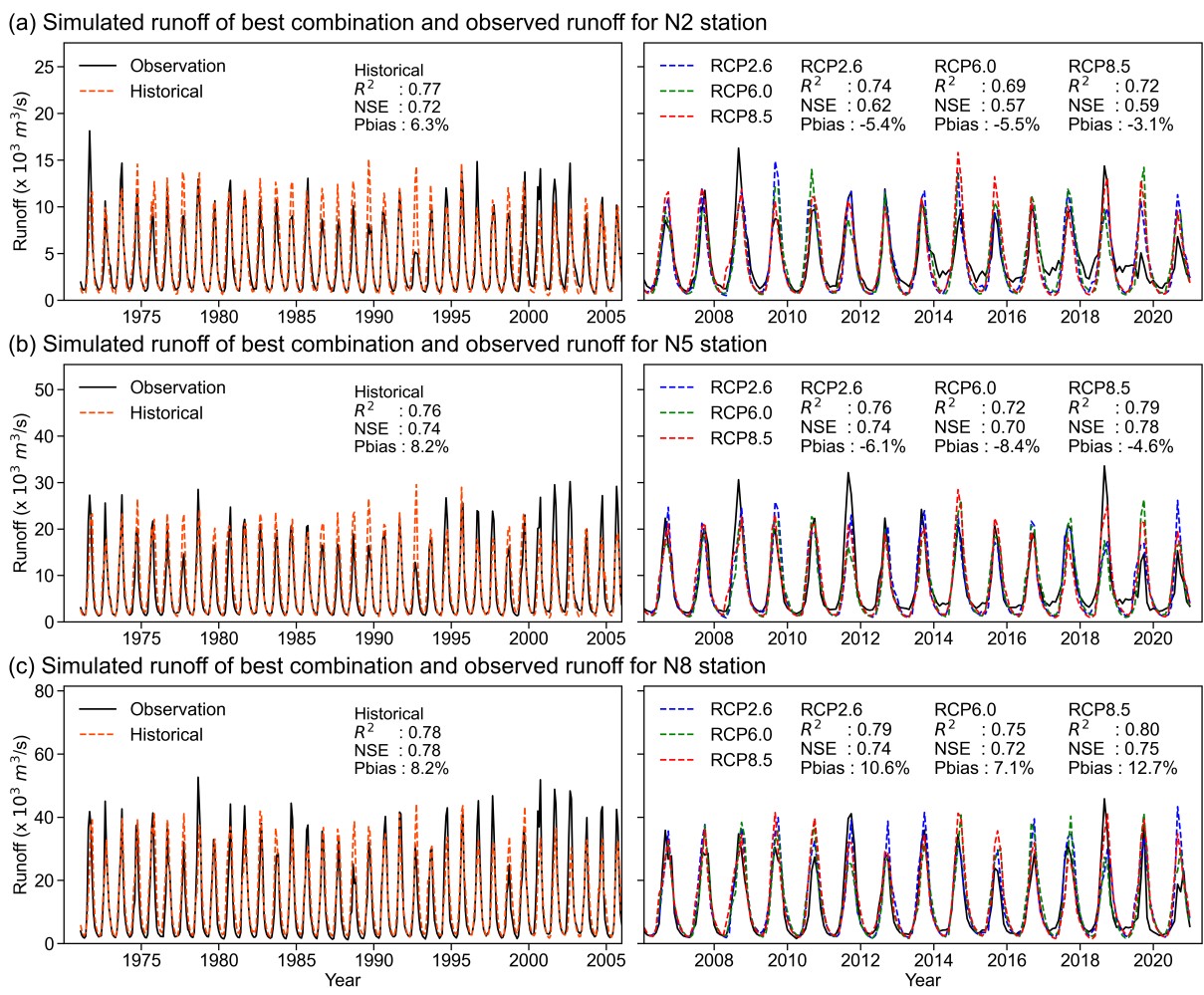

**Figure 5.** Observed and simulated monthly runoff for historical validation time periods (1971−2020) under the best combination (GCMs ensemble average and WaterGAP2). The three rows correspond to three hydrological stations (N2, N5 and N8). In each panel, the solid black line is observed runoff and the dashed coloured lines (orange, blue, green, and red) are simulated runoff under different scenarios (historical, RCP2.6, RCP6.0, and RCP8.5).

Through validation using historical data and comparing to future scenarios, it is seen that WaterGAP2 performs the best, suggesting that the model would be the best suited for the MRB runoff simulations. At the same time, the results show that the

ensemble average results of GCMs can reduce the uncertainty of future climate projections. Another comparison made is to take the historical observed runoff of a representative station in the upper, middle, and lower reaches of the MRB, and use the ensemble average of GCMs and the simulated runoff under the WaterGAP2 combination (Fig. 5). The $R^2$ and Pbias are around 0.75 and 5% respectively in the historical period and the RCPs real-time period. The above verification metrics indicate that the simulation performance of the combination at the three representative stations is satisfactory. Based on the above verification results, the GCMs ensemble average and WaterGAP2 combination to analyse the future runoff of MRB are used. It is also determined that the results of a single GCM are also of reference significance.

### 3.3 ISIMIP future projections

#### 3.3.1 Annual runoff change

MK significance tests are performed on future annual runoff changing at representative stations of the MRB (upstream: N2, midstream: N5, downstream: N8). First, the overall result (Fig. 6) is that under different RCP scenarios, the runoff in each station increases significantly (p<0.05) in MRB. Second, in terms of spatial distribution, the impact of future climate on the runoff change of the MRB becomes higher when moving from upstream to downstream. Specifically, under a given RCP scenario, the increasing rate of annual runoff at downstream stations is always higher than that at upstream stations. For example, under the RCP2.6 scenario (see the first column of Fig. 6), the annual runoff changing rate of the upstream N2 station, midstream N5 station, and downstream N8 station increased from $3.81 \pm 3.47$ m$^3$ s$^{-1}$ a$^{-1}$ to $7.40 \pm 7.41$ m$^3$ s$^{-1}$ a$^{-1}$ and final to $12.94 \pm 11.41$ m$^3$ s$^{-1}$ a$^{-1}$, respectively. There are the same results under RCP6.0 and RCP8.5 scenarios. This shows that under the future scenarios, the downstream runoff will be more affected, resulting in a higher interannual variability.

As the RCPs change (for example, from RCP2.6 to RCP6.0, and to RCP8.5), not all stations have an increasing annual runoff increment with the scenarios change. In other words, the annual runoff increasing rate under RCP8.5 are not necessarily greater than those under RCP2.6 and RCP6.0. The upstream station (N2) has the lowest runoff increasing rate ($3.81 \pm 3.47$ m$^3$ s$^{-1}$ a$^{-1}$) under the RCP2.6 scenario, and the highest runoff increasing rate ($8.72 \pm 3.93$ m$^3$ s$^{-1}$ a$^{-1}$) under the RCP8.5 scenario. At this station, precipitation and glacier snowmelt dominate the increase in runoff. The increases of precipitation and glacier snowmelt under RCP8.5 scenario is higher than those of RCP2.6 scenario and RCP6.0 scenario, which lead to the highest increasing rate of runoff at this station under RCP8.5 scenario. Different from the above N2 station results, the midstream station (N5) has the lowest runoff increasing rate ($7.40 \pm 7.41$ m$^3$ s$^{-1}$ a$^{-1}$) under the RCP2.6 scenario, while the runoff increasing rate ($10.84 \pm 7.73$ m$^3$ s$^{-1}$ a$^{-1}$) in the RCP6.0 scenario is larger than that ($10.21 \pm 7.62$ m$^3$ s$^{-1}$ a$^{-1}$) in the RCP8.5 scenario, although the precipitation increases in the RCP8.5 scenario is the highest. A possible explanation for this is that within the catchment range of these two stations, the effects of upstream glacial snowmelt and increased precipitation are attenuated by the increase in evapotranspiration caused by warming, so that the increase in runoff is also reduced in N5 station. Guan et al. (2021) reported similar results in a typical watershed in southern China. Their study points out that the rising air temperature tends to evaporate more water and offset the effect of precipitation increase to some extent, which is more pronounced at lower latitudes. The MR is a north-south river, and the latitude of the midstream N5 station is lower

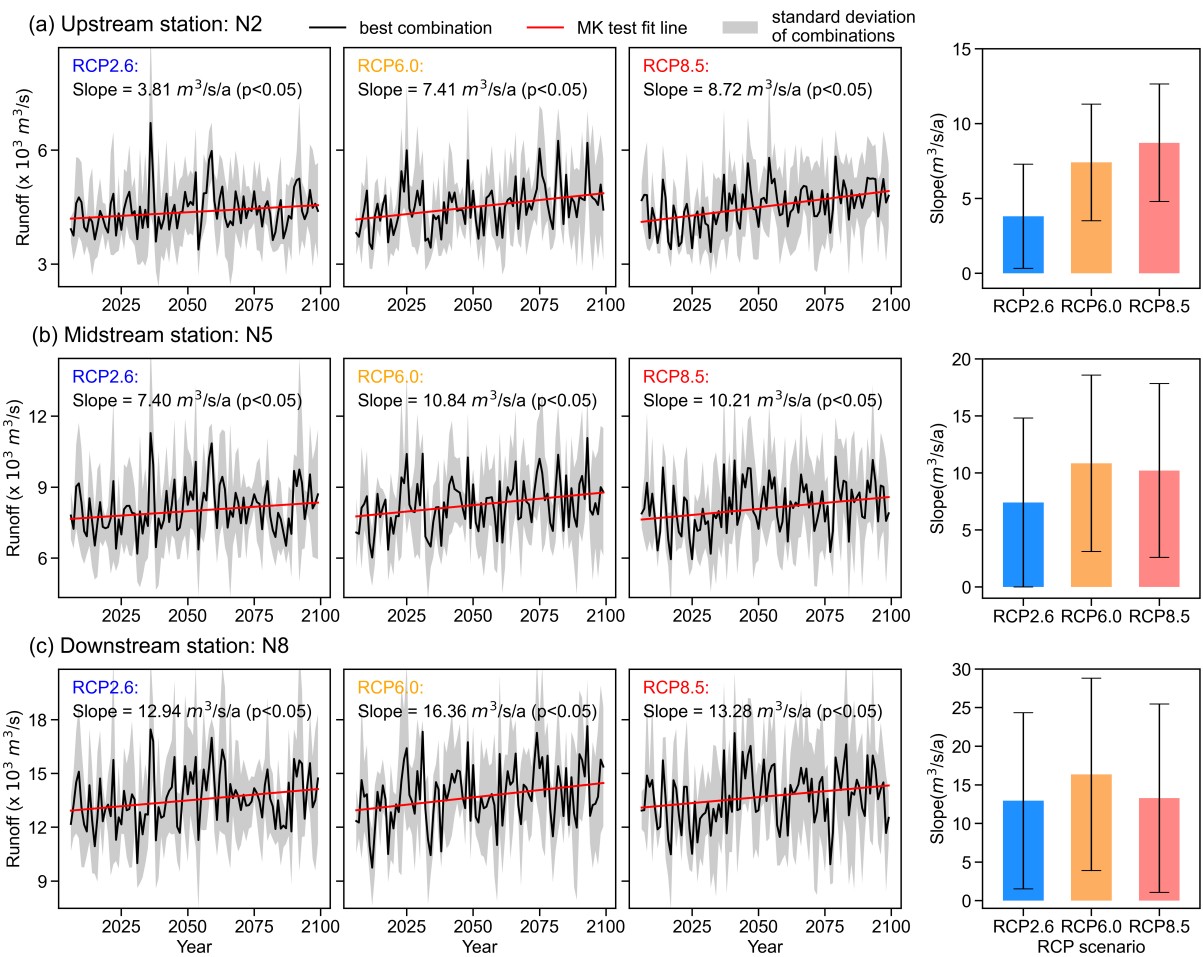

**Figure 6.** ISIMIP projections of annual discharges for 2006−2099 under different RCP scenarios. The three rows correspond to future projections of three hydrological stations (N2, N5 and N8). In each row, three panels on the left are runoff time series for three RCP scenarios (RCP2.6, RCP6.0 and RCP8.5), while one panel on the right summarize the changing rate in annual runoff under the three RCPs. Then in the right panel, the different colored bars are for the runoff changing rate under each RCP, and the error bars are the uncertainty range.

than that of the upstream N2 station. Therefore, the midstream N5 station has the highest runoff increasing rate under the RCP6.0 scenario. Consistent with the above N5 station results, there is the highest increasing rate ($16.36\pm12.44$ m$^3$ s$^{-1}$ a$^{-1}$)

of the downstream station (N8) under RCP6.0. Only 16% of the total runoff in the lower Mekong comes from China Li et al. (2017); Ruiz-Barradas and Nigam (2018). This shows that the glacial snowmelt brought by warming has limited impact on the downstream, and evapotranspiration and precipitation are the main factors affecting the downstream runoff. At the same time, at lower latitudes than the N5 station, the rising air temperature tends to increase evapotranspiration and offsets the effect of precipitation increases to a higher extent (Guan et al., 2021).

### 3.3.2 Seasonal runoff change

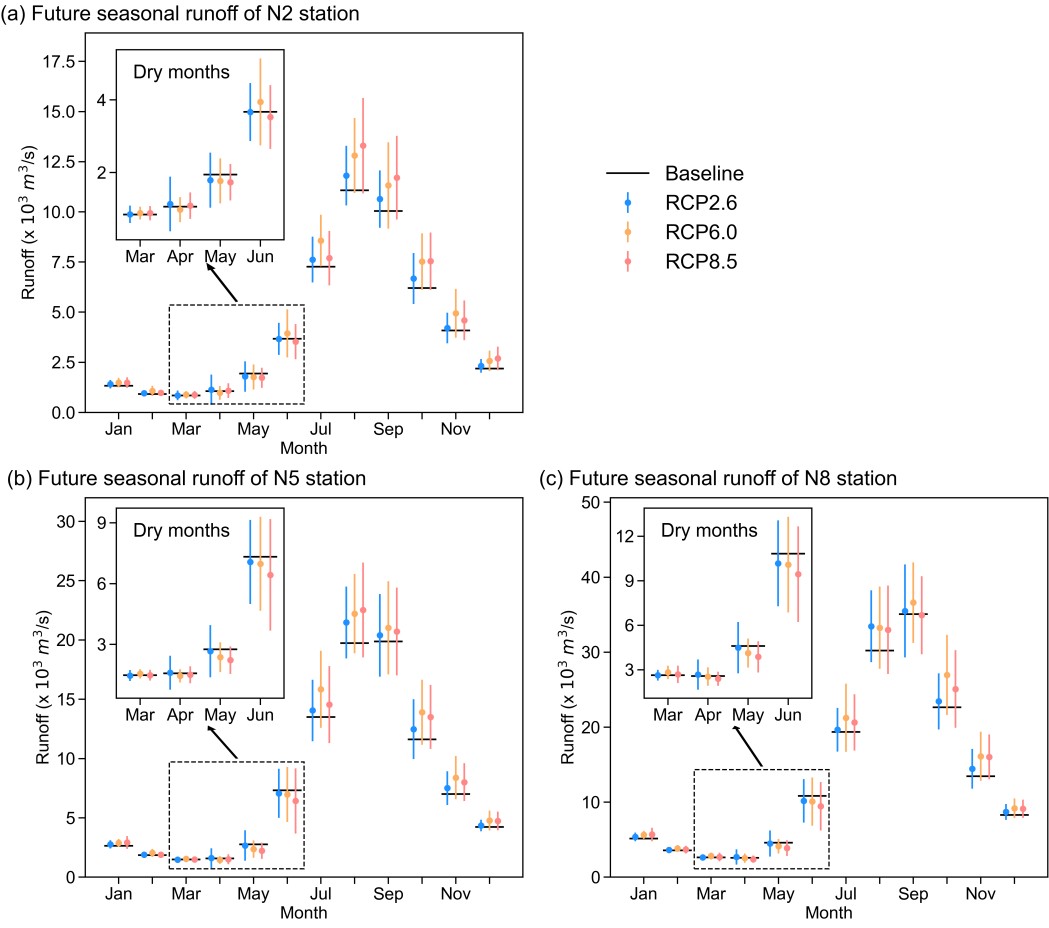

**Figure 7.** Seasonal runoff changes under different RCPs scenarios at representative hydrological stations. The three panels correspond to future projections of three hydrological stations (N2, N5 and N8). In each panel, the black horizontal line is the baseline seasonal runoff, and the three colored (blue, yellow and red) dots and vertical lines are the projected seasonal runoff and its uncertainty range under the three RCPs (RCP2.6, RCP6.0 and RCP8.5).

In order to analyse the seasonal runoff changes of MRB in different RCPs, the base period of 1991−2020 and the future period of 2070−2099 are chosen. Fig. 7 shows the intra-annual runoff change under different scenarios for representative MRB stations. Table 3 presents the percentage change in the respective runoff. Overall, the GCMs scenario ensemble results show that monthly runoff increases at representative stations, except for a decrease in May−June. The study (Hoang et al., 2016) also finds higher monthly runoff at all MRB stations, except for a slight reduction in June. In terms of time distribution within a year, the annual runoff distribution within MRB will be more uneven in the future. Specifically, the runoff increase of the representative stations is concentrated in the rainy season months, while the runoff even decreases in specific dry season months (such as May). The above results indicate that the wet months will get wetter and the dry months will get drier within MRB under the three RCP scenarios. Furthermore, this phenomenon is more prominent in the RCP8.5 scenario. For example, under this scenario, the runoff of midstream station will reduce by 23.7±23.2 % in May and increase by 16.7±23.3 % in October. In terms of the spatial distribution, the runoff changes under different RCP scenarios are particularly complex and different. For example, in October of the rainy season, the upstream station has the highest runoff increase (+ 21.5±23.0 %) under RCP8.5, while the midstream station and the downstream station have the highest runoff increase (N5: + 20.4±23.7 %; N8: + 19.7±23.7 %) under RCP6.0. On the other hand, in the dry season of May, the three representative stations all have the most prominent runoff declines (N2: - 16.5±24.3 %; N5: - 23.7±23.2 %; N8: - 18.8±22.1 %) under RCP8.5. The reasons for the different increases under different RCP scenarios are related to the latitude of the stations. The detailed reasons of above results could be seen in section 3.3.1.

## 4 Discussion

This study systematically analyses the performance and uncertainty of runoff simulations from five GHMs driven by four GCMs within the MRB during historical periods. An interesting finding is that the variability introduced by the GCMs was similar to or even greater than that introduced by the GHMs on the runoff simulation (Fig. 3 and Fig. 4). For example, in Fig. 3a, the median $R^2$ of different GHMs under the same GCM driver can differ by 0.20, but the median $R^2$ of the same GHM under different GCMs drivers can differ by more than 0.20. To reduce the variability of runoff simulation, on the one hand, we can obtain a well-performing GHM through comprehensive evaluation. In this study, three performance indicators were combined under eight hydrological stations, and WaterGAP2 (i.e., GHM) was found to have the best performance (highest $R^2$ and NSE, and lowest Pbias) under four GCM drivers in the MRB. In addition, even a good GHM has high uncertainty for future runoff projection under different GCM drivers. A feasible approach at this time should be to combine the ensemble average of runoff results from the GHM driven by different GCMs, which can help reduce the uncertainty from climate models in future projections. At the same time, the standard deviation of runoff results from the GHM driven by different GCMs can be used to quantify the uncertainty in future runoff projections.This approach gives equal weight to each GCM, often referred to as "model democracy", and has been widely used in climate impact assessments (Taylor et al., 2012; Collins et al., 2013). Another approach that can potentially reduce uncertainty is a weighting scheme that considers the performance of the GCMs (Knutti et al., 2017; Yang et al., 2017). The GCMs are weighted by different statistics in the past or present, and the

**Table 3.** Percentage of runoff change in different months under different RCP scenarios at representative station

| Station | RCP | Seasonal runoff change (%) | | | | | |
|---|---|---|---|---|---|---|---|
| | | Jan | Feb | Mar | Apr | May | Jun |
| Chiang Khan (N2) | RCP2.6 | 6.3±15.9 | 3.8±11.8 | 0.7±28.8 | 7.3±71.3 | -7.8±39.1 | -0.1±21.7 |
| | RCP6.0 | 9.2±17.4 | 12.6±27.4 | 10.2±22.4 | 7.2±38.2 | -8.9±32.0 | 6.6±32.3 |
| | RCP8.5 | 8.7±19.3 | 1.6±12.7 | 6.9±23.9 | 1.3±33.9 | -16.3±24.3 | -7.3±23.1 |
| | | Jul | Aug | Sep | Oct | Nov | Dec |
| | RCP2.6 | 4.8±15.7 | 6.6±13.5 | 5.9±14.3 | 7.5±20.5 | 2.9±18.7 | 6.1±15.7 |
| | RCP6.0 | 21.7±18.2 | 15.9±16.9 | 12.9±21.5 | 19.4±22.4 | 16.7±28.8 | 11.0±21.9 |
| | RCP8.5 | 4.1±18.4 | 18.1±21.2 | 16.1±20.7 | 21.5±23.0 | 12.4±24.2 | 19.7±26.2 |
| | | Jan | Feb | Mar | Apr | May | Jun |
| Mukdahan (N5) | RCP2.6 | 4.8±13.0 | 1.9±11.0 | -1.3±18.1 | 2.2±53.0 | -3.4±46.6 | -3.5±28.3 |
| | RCP6.0 | 7.1±12.3 | 8.7±17.8 | 7.0±16.1 | 3.9±23.1 | -11.9±27.3 | -4.6±31.7 |
| | RCP8.5 | 7.8±19.8 | -1.0±11.7 | 0.2±17.9 | -4.4±26.5 | -23.7±23.2 | -15.7±36.1 |
| | | Jul | Aug | Sep | Oct | Nov | Dec |
| | RCP2.6 | 4.1±19.2 | 8.9±15.4 | 2.6±17.6 | 7.4±21.7 | 7.3±20.3 | 2.8±11.5 |
| | RCP6.0 | 25.3±25.7 | 12.7±17.0 | 7.1±20.0 | 20.4±23.7 | 16.3±25.3 | 9.2±19.2 |
| | RCP8.5 | 7.9±24.1 | 11.4±19.8 | 5.8±18.9 | 16.7±23.3 | 12.8±22.4 | 11.4±18.3 |
| | | Jan | Feb | Mar | Apr | May | Jun |
| Stung Treng (N8) | RCP2.6 | 4.2±11.2 | 1.5±10.7 | 0.1±13.8 | 4.0±39.6 | -2.5±37.7 | -6.1±26.7 |
| | RCP6.0 | 7.2±10.2 | 7.1±12.9 | 7.9±17.3 | 5.4±25.9 | -7.5±21.8 | -6.4±29.9 |
| | RCP8.5 | 8.4±17.1 | 2.1±14.0 | 1.1±22.0 | -9.0±18.2 | -18.8±22.1 | -16.8±28.4 |
| | | Jul | Aug | Sep | Oct | Nov | Dec |
| | RCP2.6 | 1.4±15.1 | 10.7±15.8 | 1.2±17.7 | 3.5±16.5 | 7.4±19.8 | 4.9±12.8 |
| | RCP6.0 | 18.3±25.2 | 10.1±18.1 | 6.3±15.6 | 19.7±23.7 | 16.5±23.7 | 9.0±15.6 |
| | RCP8.5 | 6.8±19.4 | 5.3±18.8 | 1.1±15.1 | 10.5±22.8 | 17.4±21.9 | 9.8±14.7 |

weighting coefficients are applied to the future projections of the GCMs. However, there is a risk that a GCM that performs
poorly in the current climate may perform better when environmental conditions are beyond the contemporary range of change
(Yang et al., 2017). It is worth mentioning that a novel and promising approach to constrain uncertainties is the emergent
constraints (ECs). The EC approach consists of statistical (emergent) relationships between an observable quantity (X) in the
past or present climate and a quantity (Y) related to the future climate across GCMs (Brient, 2020; Hall et al., 2019; Schlund
et al., 2020). Combining emergent relationships with observations can potentially reduce uncertainty in future projections, and
several published ECs have shown us positive effects (Schlund et al., 2020; Shiogama et al., 2022). We encourage further
experimentation with various approaches, including those described above, to overcome the uncertainty among GCMs in the
MRB.

Another point is that under different RCPs, the interannual runoff of the three representative sites has a significant (p<0.05)
increasing trend, which is consistent with the previous relevant studies suggesting that MRB runoff would increase in the future
due to climate change (Hoang et al., 2019; Liu et al., 2022). A novel finding is that the upstream, midstream and downstream
stations in the MRB show different patterns of runoff change under three RCP scenarios. The increase in runoff at upstream
station N2 increased sequentially as the scenarios changed from RCP2.6 to RCP6.0 then to RCP8.5. The difference is that the
downstream station N8 has the highest runoff increase under the RCP6.0 scenario, while not under the RCP8.5 scenario. This
behaviour is closely related to the combined effects of temperature and precipitation on runoff under different RCP scenarios.
Specifically, at upstream stations, the synergistic effect of increased glacial meltwater and increased precipitation caused by
warming under different scenarios is greater than the effect of increased evapotranspiration caused by warming. This results in
the highest runoff increase under RCP8.5. At downstream stations, the proportion of glacier meltwater to total water volume
decreased, suggesting that its impact on total runoff was also lower. In addition, the increase in evapotranspiration due to
warming increases with decreasing downstream latitude. Under the combined effect of these factors, the runoff increases under
the RCP6.0 scenario ($16.36\pm12.44$ m$^3$ s$^{-1}$ a$^{-1}$) and is higher than that under the RCP8.5 scenario ($13.28\pm12.20$ m$^3$ s$^{-1}$
a$^{-1}$). This means that the risk of future flooding in the middle and lower reaches of the MRB is still likely to remain a high
level, even if we try to manage to stay on a moderate emissions path (i.e., RCP6.0). The novel change patterns of the upper,
middle and lower reaches explored in the study may be able to provide a scientific basis for the future implementation of local
water resource management schemes in each reach of the MRB.

Furthermore, in the far future period ($2070-2099$), the distribution of seasonal runoff within the MRB is more complex.
Despite the apparent increase in interannual runoff, water stress in the dry season would not decrease, or become more severe.
Under all RCP scenarios, runoff will decrease in future dry season months (e.g. May). Even under the RCP8.5 scenario, the
percentage of runoff reduction at representative sites in May was above 15%, reaching a maximum of 24%. This can exacerbate
water shortages during the dry season and will have particular impact on Cambodia, which, as noted above, relies on the
Mekong to fill the Ton Le Sap, and on the Mekong Delta in Vietnam. The increase in interannual runoff is mainly reflected
in the rainy months. For example, under the RCP6.0 scenario, the midstream representative station (N5) will have a runoff
increase of 25% in July. This behaviour will increase flood events in the basin, affecting human safety and normal livelihood
and economic activities. Although studies (Yun et al., 2021; Lauri et al., 2012; Wang et al., 2017) have shown that rational

reservoir operation can mitigate hydrological extremes in the basin, the management of such transboundary rivers requires closer cooperation among all the countries in the MRB. It is worth mentioning that our current study focuses on the impacts of future climate change on runoff in the MRB without quantifying dam/reservoir-induced changes in runoff, due to the limited availability of dam/reservoir data. In future work, we expect to be able to quantitatively analyse the impacts of human activities and climate change on runoff by acquiring or collecting dam/reservoir data to explore the potential of reservoir operations to mitigate the extreme hydrological events under complex future runoff change scenarios (e.g., droughts and floods).

## 5   Conclusions

From the 1970s to the present, there has been no significant (p>0.05) change in the runoff of the MRB. In the early operation stages of the reservoirs built in the 1990s, the annual runoff decreased obviously. However, the impact of the reservoir on the annual runoff after the completion of water storage is small. The ensemble-averaged results of GCMs can reduce the uncertainty of runoff simulations by different climate models. Moreover, WaterGAP2 performs the best runoff simulation at each station, with the average $R^2$, NSE and Pbias of the stations being 0.78, 0.68, and 5.5%, respectively. Based on these evaluation results, the WaterGAP2 runoff simulation has been used in the MRB to analyse runoff changes under future scenarios.

Under the RCP scenarios, the future inter-annual runoff of the MRB increases significantly (p<0.05). Notably, the upper and lower reaches of the MRB show different patterns of runoff change under three RCP scenarios, associated with the combined effects of temperature and precipitation on runoff for each reach. Under the RCP6.0 scenario, the MRB has the highest increase in interannual runoff. Seasonal changes in annual runoff in the MRB under future climate are more complex. Under the RCP2.6 and RCP6.0 scenarios, the runoff of the MR during the rainy season will increase, and the increase in RCP6.0 is higher than that in RCP2.6. The changes of runoff in the dry season are relatively stable under the two scenarios. However, the seasonal runoff changes in the MRB under the RCP8.5 scenario are extremely complex. The specific performance of the ensemble average of the GCMs and the WaterGAP2 combination suggests that the dry season will become drier, the rainy season wetter and the distribution of water resources over the year more uneven. Overall, this study provides novel insights for future runoff projections from a whole river system perspective and may be able to offer a scientific basis for the future implementation of water resource management schemes in the MRB.

*Code and data availability.*  The observed runoff data can be obtained by contacting the Mekong River Commission (MRC; https://portal. mrcmekong.org/). All model result data used are openly available from the Inter-Sectoral Impact Model Intercomparison Project (ISIMIP; https://data.isimip.org/search/). The codes for data processing and result visualization are available upon request from the corresponding author.

*Author contributions.*  AC conceived and designed the study. AC and CW collected the data, performed the analysis, and drafted the original manuscript. SL, LL, XS, JM, YZ, and AC made suggestions and revised the manuscript. All authors contributed to the discussion.

*Competing interests.* AC is a member of the editorial board of Earth System Dynamics. The peer-review process was guided by an independent editor, and the authors also have no other competing interests to declare.

*Acknowledgements.* We acknowledge the Mekong River Commission (MRC) for providing the observed runoff data. The research was supported by the National Natural Science Foundation of China (92047302 and 51961125203). X. Shi and J. Mao were supported by the Reducing Uncertainties in Biogeochemical Interactions through Synthesis and Computing Scientific Focus Area (RUBISCO SFA) project funded through the Earth and Environmental Systems Sciences Division of the Biological and Environmental Research Office in the US Department of Energy (DOE) Office of Science. Oak Ridge National Laboratory is supported by the Office of Science of the DOE under Contract No. DE-AC05-00OR22725.

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
