# Peer review of "Historical and projected future runoff over the Mekong River Basin"

_EGUsphere, 2023_

## Author Comment (AC1)

**Reviewer 1**

Thank you for inviting me to review the paper: "Historical and Projected Future Runoff over the Mekong River Basic" by Wang et al.

This is a useful and important paper. It is useful because it builds links between the detailed hydrological modelling community and those developing GCMs. Too often, climate researchers consider a relatively basic land surface model in a GCM as sufficient – but in reality, something much better is needed to help understand future flooding impacts. The paper is important because, as the authors state, 65 million rely on the Mekong River for access to water.

The analysis is well-considered and thorough. My only concern with the paper is that there needs to be better wording and removing ambiguity in places. All of this can be easily rectified in the generation of the new paper version (and I am happy to re-review any revisions).

Response: We appreciate the reviewer's positive feedback and helpful comments, which are highly helpful for us to improve the manuscript. Please kindly find below our detailed responses to each of your comments. Texts in blue are our responses to the comments, while those in red are revisions of the manuscript.

Below are illustrative examples, but please check through the entire manuscript.

P2, Line 48. "However, these studies do not systematically analyse….". Is this suggesting that substantial errors could occur with good GHMs, should there be major biases in GCMs (so a GCM+GHM combination fails, even if the GHM is good).

Response: Yes, as we mentioned in Line 26 of the manuscript, "*Different GCMs use distinct representations of the climate system, leading to "climate model structural uncertainty" (Gosling and Arnell, 2011).* ". We have added this information following the sentence in Line 48 in the revised manuscript:

"*However, these studies do not systematically analyze the runoff simulation results of long-term historical periods (including the historical period of historical scenarios and the real-time period of representative concentration pathways (RCP) scenarios, i.e. from the start simulation year of the RCPs to now pre-2023, for which observed runoff data are available.) under different GCM-GHM combinations. Such an analysis is meaningful and urgent to potentially assess and reduce the uncertainty/bias of runoff simulations introduced by both GCMs and GHMs (Kingston et al., 2011; Hoang et al., 2016) .*"

P2, Line 49. The wording here is clumsy. Maybe something like: "and the simulated years at the beginning of the RCP scenarios, which are now pre-2023 and for which runoff data exists."

Response: Thanks for the helpful suggestion. We have changed this sentence in the revised manuscript:

"*However, these studies do not systematically analyse the runoff simulation results of long-term historical periods (including the historical period of historical scenarios and the real-time period of representative concentration pathways (RCP) scenarios, i.e. from the start simulation year of the RCPs to now pre-2023, for which observed runoff data are available.) under different GCM-GHM combinations.*"

P3, Line 58. Please state what ISI-MIP is. Are the GCM-GHM combinations already calculated in ISI-MIP, or is that database just GCM output? Are any of the outputs from ISI-MIP already bias corrected?

Response: The Inter-Sectoral Impact Model Intercomparison Project (ISIMIP) is a community-driven modelling effort and offers a framework for comparing climate impact projections in different sectors and at different scales (Warszawski et al., 2014). Specifically, the ISIMIP 2b scenarios are designed to elicit the contribution of climate change to impacts arising from low-emissions climate-change scenarios (Frieler et al., 2017). You are right that all the GCM-GHM combinations already calculated in ISIMIP 2b, and the results of runoff simulations of five GHMs forced by four GCMs are all derived from the experimental data of the global water sector in ISI-MIP2b. Here, all GCM output meteorological forcing have been bias adjusted. These adjusted meteorological outputs have been collected in the EWEMBI dataset and used as meteorological forcing inputs for all the GHMs in the ISI-MIP2b (Frieler et al., 2017). We have added above information on the ISIMIP project in Section 2.3 in the revised manuscript:

 "*2.3 Climate projections and hydrological models*
*ISI-MIP is a community-driven modelling effort and offers a framework for comparing climate impact projections in different sectors and at different scales(Warszawski et al., 2014). In the ISI-MIP, the ISIMIP 2b scenarios are designed to elicit the contribution of climate change to impacts arising from low-emissions climate-change scenarios (Frieler et al., 2017).* The global climate models (GCMs) selected for this study are derived from the ISIMIP 2b protocol, which provides four GCMs from CMIP5 and three emission scenarios (i.e., RCP2.6, RCP6.0 and RCP8.5). … All GHMs operate under the meteorological drive of the four GCMs, and the ensemble-averaged results of the GCMs are also evaluated due to the variability of the GCMs and the uncertainty of climate change. The above runoff*

*simulation results of five GHMs forced by four GCMs are all derived from the experimental data of the global water sector in ISIMIP2b.*"

P3, Line 72. Please clarify why you would use the MK test, and not the standard statistical test of whether a regression line is statistically significant.

Response: The Mann-Kendall (MK) test (Mann, 1945; Kendall, 1948) is a rank-based non-parametric method. Compared to parametric tests (e.g., regression coefficient test), non-parametric tests (e.g., the MK test) have no requirements of homoscedasticity or prior assumptions on the distribution of the data sample (Bihrat and Bayazit, 2003) and are less sensitive to outliers (Hamed, 2007). As the MK test statistic is determined by the ranks and sequences of time series rather than the original values, it is robust when dealing with non-normally distributed data, which are commonly encountered in hydrometeorological time series (Wang et al., 2020). We will provide a brief explanation of the choice of the MK test in the revised manuscript.

P5, Line 91. Please state where the GHMs come from. Is it a database such as ISI-MIP. State here that information, and also in the caption of Table 2 (Also, please check the current Caption of Table 2 – it looks wrong, referring to eight hydrological stations).

Response: The runoff simulations results of five GHMs forced by four GCMs were all derived from the experimental data of the global water sector in ISI-MIP2b, which is openly available on ISIMIP protocol (https://data.isimip.org/search/product/). Meanwhile, thanks for your reminder, we have checked the caption of Table 2 and added the information of GHMs sources in the revised manuscript:

"*Table 2:  Basic information of the GHMs in the ISIMIP2b Global Water program. The runoff simulation results of the GHMs forced by different GCMs are all derived from the ISIMIP protocol (https://data.isimip.org/search/product/).*"

There is one technical issue. Could the authors describe if there is any bias correction undertaken e.g. of the ESMs? To my knowledge, some ESMs in the "MIPs" are corrected. If so, are these used – because they should reduce climate uncertainty/errors in any GCM+GHM projections of the contemporary period?

Response: As mentioned earlier, the meteorological forcing from the GCMs in ISI-MIP2b have performed bias adjustment to reduce climate uncertainty/error in future projections. For the detailed description of the bias adjustment, please refer to Frieler et al. (2017).

P6, Table 2 – as noted elsewhere, captions appear overly succinct. I often see people give talks where diagrams and tables are extracted from papers, so if they can be more complete (i.e. with essential details in captions), then this is very helpful.

Response: Thanks for your reminder and suggestion. We have changed the caption of Table 2 to make it more complete in the revised manuscript:

"*Table 2:*  *Basic information of the GHMs in the ISIMIP2b Global Water program. The runoff simulation results of the GHMs forced by different GCMs were all derived from the ISIMIP protocol ([https://data.isimip.org/search/product/](https://data.isimip.org/search/product/))."*

P7, Figure 2 – One possibility to avoid the repeated words "insignificant change" is to give the p-value for the regression. Then where it is significant (e.g. p < 0.05), mark with a star symbol. Something like that… Also, again, expand the caption slightly. For instance, state, "Eight hydrological stations are numbered N1-N8, with their locations presented in the map of Figure 1".

Response: Thanks for your helpful suggestions. We have checked the Figure 2 and added p-value where significance tests are involved. We have also changed the caption of Figure 2 in the revised manuscript:

[Figure]

*Figure 2: The results of the MK trend test in historical (1971-2020) runoff over the*

*eight hydrological stations. Eight hydrological stations are numbered N1-N8, with their locations presented in the map of Figure 1.*

P7, Figure 2. Although very obvious, please put the word "Year" under panels (f), (g) and (h).

Response: Thanks for your helpful suggestions. We have revised Figure 2 accordingly in the revised manuscript, which can be found in the response to the previous comment.

P8. Table 3. The table is nice, but isn't all the information in Figures 1 and 2?

Response: Thanks for your helpful suggestions. The purpose of Table 3 is to provide the information of significant test of the changing trend. Considering the redundancy of the information, we have added significance test information to Figure 2 and then deleted Table 3. The revised Figure 2 can be found in the response to the previous comment.

Figure 3. Again, please make the caption much more informative. Something like "Figure 3. Performance of all combinations of GCMs and GHMs. The three rows correspond to three performance matrices……. In each row, each panel is for a different GCM, as annotated. Then in each panel, the different colors are for each GHM, as marked under each row……"

Response: Thanks for your helpful suggestions. We have changed the caption of Figure 3 in the revised manuscript according to the above suggestions:

[Figure]

"*Figure 3. Performance of all combinations of GCMs and GHMs during historical (1971-2005) periods. The three rows correspond to three performance matrices ($R^2$, NSE and Pbias) of all GCM-GHM combinations at the eight hydrological stations. In each row, each panel is for a different GCM, as annotated. Then in each panel, the different colors are for each GHM, as marked under each row.*"

Furthermore, we have also changed the captions of the remaining figures to make them more informative in the revised manuscript.

Figure 3. Although it is important to make captions informative, placing a result there is not usual practice. So please reconsider the words "WaterGap2 has the best performance compared to other models", and should they be in a caption?

Response: Thanks for your helpful suggestions. We have removed this sentence from the captions of Figure 3. The revised the captions of Figure 3 can be found in the response to the previous comment.

Figure 3. And on the same point above, statements such as "…best performance compared to other models" requires very careful quantification. Is it the best performance when WaterGap2 is driven by a specific GCM? Is it the best for one statistical metric or many?

Response: Thanks for your helpful suggestions. Specifically, we have quantitatively compared three performance metrices of WaterGap2 and other hydrological models under forcing from each of the four GCMs. It was found that WaterGap2 has the highest $R^2$ and NSE and the lowest Pbias than other GHMs under any *same* GCM forcing. This shows that despite the uncertainty between different GCMs, the simulation of the MRB runoff by WaterGap2 has better performance and reliability than other models. We have clarified this point in Line 141 of the revised manuscript:

Line 141: "*Among these GHMs,  WaterGap2 has the highest $R^2$ and NSE and the lowest Pbias than others under the same GCM forcing.*"

In the Conclusions, critical is consistency between direct drivers of runoff change and changes imposed by raised GHGs and related altered rainfall patterns. Here, the impression is that the former is small (i.e. "However, the impact of the reservoir on the annual runoff after the completion of water storage is small"). Elsewhere in the paper, there is the suggestion that humans have impacted – directly – runoff strongly. I still think it would be useful to have a summary statistic that is some sort of ratio between historical direct change and the impact of raised GHGs on runoff. This should be easy to do, as all the numerical values to build such a single comparison statistic are calculated at different points within the manuscript.

Response: Thanks for your helpful suggestions. The ratio you mention, comparing historical direct changes and future climate impacts on runoff, is undoubtedly meaningful and attractive. In our current manuscript, the historical direct changes are calculated from observed runoff and are also influenced by both historical climate change and historical human activities. Direct comparisons between the combined effects of historical climate change and human activities on runoff and the separate effects of future climate change on runoff may lead to misleading conclusions because the interactions of climate change and human activities on runoff over historical periods are usually complex and unclear. With sufficient reservoir/dam-related data, we can hopefully separate the effects of historical climate change from the effects of historical human activities and compare them to the effects of future climate change. In the absence of reservoir/dam data, our current work only calculates direct historical changes and does not compare them to future climate change impacts. In the revised manuscript, we will include a more thorough discussion on this point, and- highlight the importance of future efforts to record or collect local reservoir/dam data to further explore the relationship between human activities impacts and climate change impacts.

Small things – here, for Abstract but may be representative elsewhere

Abstract: These need to avoid ambiguity, as often read in isolation by a reader in a hurry. Hence please:

(1)    tighten line 7 and explain the difference between how "four GCMs" and "five GHMs" are used. Are they operated independently e.g. raw runoff output from the GCMs are used – while the GHMs are forced with known near-surface meteorological drivers? The next sentence, however, talks about "best simulation combination", so state this is 4 x 5 simulations – all combinations of GCMs and GHMs (I realise the main body of the paper makes this clearer, but such very basic information should be in an Abstract).

Response: Thanks for your helpful suggestions. We apologize for any confusion caused by the wording here. Here we use the runoff simulation output of five GHMs driven by four GCMs, which is the combination of GCM and GHM described in the manuscript. To avoid confusion, we have clarified in Line 7 in the revised manuscript:

"*With these runoff data, we then evaluate the runoff simulation performance of  five global hydrological models (GHMs) forced by four global climate models (GCMs) under the ISI-MIP project.*"

(2)    Line 11. State what "WaterGap2" is (i.e. GHM).

Response: Thanks for your helpful suggestions. We have clarified in Line 11 in the revised manuscript:

"* The ensemble-averaged result of WaterGap2 (i.e., GHM) forced by four GCMs has the best runoff simulation performance.*"

(3)    The Abstract presents two lines of investigation but does not bring them together in a coherent way. One direction is that for the contemporary period, it is dams and reservoirs that have had the biggest effect on runoff. However, when describing the future based on RCPs, runoff is described as "projected to increase significantly". The question is then whether future changes caused by climate change are bigger than current changes caused by dams/reservoirs? (See similar comment above)

Response: Thanks for your helpful suggestions. Quantitative comparisons of future changes caused by climate change and current changes caused by dams/reservoirs are meaningful and attractive. Unfortunately, we currently lack sufficient dams/reservoirs data to quantify the current impact of dams/reservoirs on runoff (see previous response

above). Our current manuscript focuses on qualitative analysis of the current impact of dams/reservoirs on runoff and quantitative analysis of the future impact of climate change on runoff. In future work, we expect to be able to quantitatively analyze the impact of human activities and climate change on runoff by acquiring or collecting reservoir data.

(4)     Line 13. Is "increase significantly" a formal statistical statement, and should there be a p-value?

Response: Yes, here is a formal statistical statement where a p-value is required. We have clarified this in Line 13 in the revised manuscript:

"*Under representative concentration pathways (RCPs, i.e., RCP2.6, RCP6.0 and RCP8.5), runoff of the MR is projected to increase significantly ($p<0.05$, from 3.81 $m^3$ $s^{-1}$ $a^{-1}$ to 16.36 $m^3$ $s^{-1}$ $a^{-1}$).*"

To be more rigorous, we have added p-values to all formal statistical statements and replaced "significant" with synonyms for informal statistical statements throughout the current manuscript. All these revisions are included in the revised manuscript. The significance level used in this study is clarified in Section 2.2 in the revised manuscript:

"*The null hypothesis in this test is that there is no significant trend in the time series at the significance level of p. If $|U| \geq U_{p/2}$, where $U_{p/2}$ is the critical value of the standard normal distribution with a probability exceeding p/2, then the null hypothesis is rejected, namely the trend is significant (Guan et al., 2021). This study adopts the significance level of 0.05, which means that there is a significant trend of change when the p-value is less than 0.05.*"

(5)     Line 13. Actual values are given here (units of m^3 s^-1 a^-1). Similar to the comments above, how large are the 3.81 – 16.36 numbers compared to the effects of dams/reservoirs? And how large are these numbers compared to background contemporary flows. Would a simple statistical value help?

Response: Thanks for your helpful suggestions. Considering the quantitative effects of dams/reservoirs are not available, we have only added the ratio of the actual values to their background contemporary. We have clarified in Line 13 in the revised manuscript:

"*Under representative concentration pathways (RCPs, i.e., RCP2.6, RCP6.0 and RCP8.5), runoff of the MR is projected to increase significantly ($p<0.05$), e.g., 3.81 $m^3$ $s^{-1}$ $a^{-1}$ ( 9% increase in 100 years) at the upstream station under RCP2.6 and 16.36 $m^3$ $s^{-1}$ $a^{-1}$ (13% increase in 100 years) at the downstream station under RCP6.0.*"

**References:**

Bihrat, Ö. and Bayazit, M.: The power of statistical tests for trend detection, 27, 247-251, 2003.

Frieler, K., Lange, S., Piontek, F., Reyer, C. P. O., Schewe, J., Warszawski, L., Zhao, F., Chini, L., Denvil, S., Emanuel, K., Geiger, T., Halladay, K., Hurtt, G., Mengel, M., Murakami, D., Ostberg, S., Popp, A., Riva, R., Stevanovic, M., Suzuki, T., Volkholz, J., Burke, E., Ciais, P., Ebi, K., Eddy, T. D., Elliott, J., Galbraith, E., Gosling, S. N., Hattermann, F., Hickler, T., Hinkel, J., Hof, C., Huber, V., Jägermeyr, J., Krysanova, V., Marcé, R., Müller Schmied, H., Mouratiadou, I., Pierson, D., Tittensor, D. P., Vautard, R., van Vliet, M., Biber, M. F., Betts, R. A., Bodirsky, B. L., Deryng, D., Frolking, S., Jones, C. D., Lotze, H. K., Lotze-Campen, H., Sahajpal, R., Thonicke, K., Tian, H., and Yamagata, Y.: Assessing the impacts of 1.5 °C global warming – simulation protocol of the Inter-Sectoral Impact Model Intercomparison Project (ISIMIP2b), Geoscientific Model Development, 10, 4321-4345, 10.5194/gmd-10-4321-2017, 2017.

Gosling, S. N. and Arnell, N. W.: Simulating current global river runoff with a global hydrological model: model revisions, validation, and sensitivity analysis, Hydrological Processes, 25, 1129-1145, 10.1002/hyp.7727, 2011.

Guan, X., Zhang, J., Bao, Z., Liu, C., Jin, J., and Wang, G.: Past variations and future projection of runoff in typical basins in 10 water zones, China, Sci Total Environ, 798, 149277, 10.1016/j.scitotenv.2021.149277, 2021.

Hamed, K. H.: Improved finite-sample Hurst exponent estimates using rescaled range analysis, 43, 2007.

Hoang, L. P., Lauri, H., Kummu, M., Koponen, J., van Vliet, M. T. H., Supit, I., Leemans, R., Kabat, P., and Ludwig, F.: Mekong River flow and hydrological extremes under climate change, Hydrology and Earth System Sciences, 20, 3027-3041, 10.5194/hess-20-3027-2016, 2016.

Kendall, M. G.: Rank correlation methods, 1948.

Kingston, D. G., Thompson, J. R., and Kite, G.: Uncertainty in climate change projections of discharge for the Mekong River Basin, Hydrology and Earth System Sciences, 15, 1459-1471, 10.5194/hess-15-1459-2011, 2011.

Mann, H. B.: Nonparametric tests against trend, 245-259, 1945.

Wang, F., Shao, W., Yu, H., Kan, G., He, X., Zhang, D., Ren, M., and Wang, G.: Re-evaluation of the Power of the Mann-Kendall Test for Detecting Monotonic Trends in Hydrometeorological Time Series, Frontiers in Earth Science, 8, 10.3389/feart.2020.00014, 2020.

Warszawski, L., Frieler, K., Huber, V., Piontek, F., Serdeczny, O., and Schewe, J.: The Inter-Sectoral Impact Model Intercomparison Project (ISI-MIP): project framework, Proc Natl Acad Sci U S A, 111, 3228-3232, 10.1073/pnas.1312330110, 2014.

---

## Author Comment (AC2)

**Reviewer 2**

General Comments:

The paper by Wang et al. provides a comprehensive analysis of historical and projected future runoff in the Mekong River Basin (MRB). The authors examined the runoff using four Global Climate Models (GCMs) and five Global Hydrological Models (GHMs) sourced from the Inter-Sectoral Impact Model Intercomparison Project (ISIMIP) 2b. These models were applied to data from eight gauge stations across MRB, considering three Representative Concentration Pathways (RCP2.6, RCP6.0, RCP8.5). The results indicate that while the annual runoff in the basin has remained relatively stable since 1971, significant increases are projected under the various RCPs.

In general, the paper is a strong fit for the journal, and its subject matter aligns well with the journal's scope. The study's focus on runoff in the MRB is of paramount importance for the region, making the findings particularly valuable. Additionally, the methodology employed in the research is both standard and widely accepted within the field.

However, one aspect that requires improvement is the clarity regarding the paper's contribution or novelty. The authors should explicitly highlight what sets their work apart from the numerous previous studies on future runoff and streamflow projections in the MRB.

Another crucial aspect overlooked by the authors is the estimation of uncertainty associated with their future projections. In studies of this nature, accounting for uncertainty is of utmost importance and cannot be overlooked. Although the authors attempted to mitigate uncertainty by using the ensemble-average approach, it falls short in providing a comprehensive estimate of the associated uncertainty. Addressing and quantifying uncertainty in their projections would add significant value to the paper's findings and enhance the overall robustness of their research.

Response: We thank the reviewer for the thoughtful comments and suggestions which we believe are greatly helpful to improve the manuscript. In the revision, the paper's contribution/novelty has been emphasized appropriately. Meanwhile, we further quantified the uncertainty in future projections. Please kindly find below our detailed responses to each comment. Texts in blue are our responses to the comments, while those in red are revisions of the manuscript.

Specific comments:

1.      Does the paper address relevant scientific questions within the scope of ESD?

Yes. The paper provides a comprehensive analysis of historical and future runoff in MRB (hydrosphere) under climate change (global change). The paper aims to quantify the impact of climate change on runoff, and thus improving our understanding of hydrological system behavior to global changes. Thus, I believe the paper aligns well with the scope of Earth System Dynamics (ESD).

Response: Thanks for your positive feedback and comment.

2.	Does the paper present novel concepts, ideas, tools, or data?

No. While the results presented in the paper are undoubtedly interesting and useful, it is worth noting that there have been several previous studies conducted in the Mekong River Basin (MRB) with similar objectives. For example, https://doi.org/10.3390/w12061556 and https://doi.org/10.1016/j.eng.2021.06.026. Additionally, the data and methodology adopted by the authors is also widely used. Therefore, it is difficult to understand the novelty of the work.

Response: Thanks for your helpful comments. As mentioned in Lines 42-48 of the current manuscript, there have been previous studies on MRB runoff changes (Kingston et al., 2011; Yun et al., 2020; Wang et al., 2021; Lee et al., 2020; Liu et al., 2022). This work distinguishes from those previous studies in several important methodological perspectives. (1) Model ensemble: Previous studies generally focused on evaluating and predicting runoff simulations with a single hydrological model driven by different GCMs or various hydrological models driven by a single GCM (Kingston et al., 2011; Yun et al., 2020; Wang et al., 2021). This work is distinctive in the use of *both various GHMs runoff simulations and driven by different GCMs*. This approach allows us to systematically analyze the performance and uncertainty of different models and separate sources of uncertainty from hydrological models and climate models. By evaluating and selecting the best GCM-GHM combination, this approach is promising to increase our confidence in the projection of the future MRB runoff. (2) Validations across the river system: Previous studies analyzing future runoff projections for the MRB under different RCP scenarios usually focused on a single station/reach (Shrestha et al., 2016; Lee et al., 2020). These studies may overlook that fact that the upper and lower reaches of the MRB may have different change patterns under different RCP scenarios. For example, as the future emissions path changes from low to high (from RCP2.6 to RCP6.0 and then to RCP8.5) in the MRB, the runoff changing rate of upstream stations may increase sequentially, but that of downstream stations may first increase and then decrease. Our work provides novel insights from the whole river system perspective. (3) Testing model projections using the 2006-2020 data: ISIMIP2b projections are published before 2006 so its future projections include the period 2006−2020. Our work is unique in using this

period, which now has real-time/world observations, to test model projections. This could increase the reliability of the simulation for the further future period.

Following your comments, we will provide clearer explanation on these novel points of our research that is distinctive from previous studies. For example:

On Line 48, we change the text to: "*However, these studies do not systematically analyze the runoff simulation results of long-term historical periods (including the historical period of historical scenarios and the real-time period of representative concentration pathways (RCP) scenarios, i.e. from the start simulation year of the RCPs to now pre-2023, for which observed runoff data are available.) under different GCM-GHM combinations. Such an analysis is meaningful and urgent to potentially assess and reduce the uncertainty/bias of runoff simulations introduced by both GCMs and GHMs (Kingston et al., 2011; Hoang et al., 2016).*"

On Line 60, we change to text to: "*(3) Finally,  we comprehensively analyze the future runoff change patterns (including annual runoff and seasonal runoff) in the upper, middle and lower reaches of the MRB under future RCP scenarios based on the best GCM-GHM combination.*"

3.      Are substantial conclusions reached?

Yes, based on the GHM simulation, the authors were able to reach at the conclusion that runoff in the MRB is projected to increase under climate change scenarios. However, the paper has not presented estimate of uncertainty associated with the projections.

Response: Thanks for your positive feedback and helpful comment. In the revised manuscript, we have quantified the uncertainty of future projections using the standard deviation of the runoff results from GHM (e.g., WaterGAP2) under four GCMs drives. We have presented the method for quantifying uncertainty in the revised manuscript:

Section 2.3 Climate projections and hydrological models
  "*All GHMs operate under the meteorological drive of the four GCMs, and the ensemble-averaged results of the GCMs are also evaluated due to the variability of the GCMs and the uncertainty of climate change. The standard deviation of the outputs of the GHM driven by four GCMs is used to quantify the uncertainty from the GCMs.*"

Furthermore, the uncertainty range has been added to Figures 6-7 and Table 4 in the revised manuscript. Meanwhile, the uncertainty range has also been marked in the text in the revised manuscript when quantifying future runoff projections.

[Figure]

"*Figure 6: ISIMIP projections of annual runoff for 2006-2099 under different RCP scenarios. The three rows correspond to future projections of three hydrological stations (N2, N5 and N8). In each row, three panels on the left are runoff time series for three RCP scenarios (RCP2.6, RCP6.0 and RCP8.5), while one panel on the right summarize the changing rate in annual runoff under the three RCPs. Then in the right panel, the different colored bars are for the runoff changing rate under each RCP, and the error bars are the uncertainty range.*"

[Figure]

"*Figure 7: Seasonal runoff changes under different RCPs scenarios at representative hydrological stations. The three panels correspond to future projections of three hydrological stations (N2, N5 and N8). In each panel, the black horizontal line is the baseline seasonal runoff, and the three colored (blue, yellow and red) dots and vertical lines are the projected seasonal runoff and its uncertainty range under the three RCPs (RCP2.6, RCP6.0 and RCP8.5).*"

*Table 4: Percentage of runoff change in different months under different RCP scenarios at three representative stations.*

| Station | RCP | Seasonal runoff change (%) | | | | | |
|---|---|---|---|---|---|---|---|
| | | Jan | Feb | Mar | Apr | May | Jun |
| Chiang Khan (N2) | RCP2.6 | 6.3±15.9 | 3.8±11.8 | 0.7±28.8 | 7.3±71.3 | -7.8±39.1 | -0.1±21.7 |
| | RCP6.0 | 9.2±17.4 | 12.6±27.4 | 10.2±22.4 | 7.2±38.2 | -8.9±32.0 | 6.6±32.3 |
| | RCP8.5 | 8.7±19.3 | 1.6±12.7 | 6.9±23.9 | 1.3±33.9 | -16.3±24.3 | -7.3±23.1 |
| | | Jul | Aug | Sep | Oct | Nov | Dec |
| | RCP2.6 | 4.8±15.7 | 6.6±13.5 | 5.9±14.3 | 7.5±20.5 | 2.9±18.7 | 6.1±15.7 |
| | RCP6.0 | 21.7±18.2 | 15.9±16.9 | 12.9±21.5 | 19.4±22.4 | 16.7±28.8 | 11.0±21.9 |
| | RCP8.5 | 4.1±18.4 | 18.1±21.2 | 16.1±20.7 | 21.5±23.0 | 12.4±24.2 | 19.7±26.2 |
| | | Jan | Feb | Mar | Apr | May | Jun |
| Mukdahan (N5) | RCP2.6 | 4.8±13.0 | 1.9±11.0 | -1.3±18.1 | 2.2±53.0 | -3.4±46.6 | -3.5±28.3 |
| | RCP6.0 | 7.1±12.3 | 8.7±17.8 | 7.0±16.1 | 3.9±23.1 | -11.9±27.3 | -4.6±31.7 |
| | RCP8.5 | 7.8±19.8 | -1.0±11.7 | 0.2±17.9 | -4.4±26.5 | -23.7±23.2 | -15.7±36.1 |
| | | Jul | Aug | Sep | Oct | Nov | Dec |
| | RCP2.6 | 4.1±19.2 | 8.9±15.4 | 2.6±17.6 | 7.4±21.7 | 7.3±20.3 | 2.8±11.5 |
| | RCP6.0 | 25.3±25.7 | 12.7±17.0 | 7.1±20.0 | 20.4±23.7 | 16.3±25.3 | 9.2±19.2 |
| | RCP8.5 | 7.9±24.1 | 11.4±19.8 | 5.8±18.9 | 16.7±23.3 | 12.8±22.4 | 11.4±18.3 |
| | | Jan | Feb | Mar | Apr | May | Jun |
| Stung Treng (N8) | RCP2.6 | 4.2±11.2 | 1.5±10.7 | 0.1±13.8 | 4.0±39.6 | -2.5±37.7 | -6.1±26.7 |
| | RCP6.0 | 7.2±10.2 | 7.1±12.9 | 7.9±17.3 | 5.4±25.9 | -7.5±21.8 | -6.4±29.9 |
| | RCP8.5 | 8.4±17.1 | 2.1±14.0 | 1.1±22.0 | -9.0±18.2 | -18.8±22.1 | -16.8±28.4 |
| | | Jul | Aug | Sep | Oct | Nov | Dec |
| | RCP2.6 | 1.4±15.1 | 10.7±15.8 | 1.2±17.7 | 3.5±16.5 | 7.4±19.8 | 4.9±12.8 |
| | RCP6.0 | 18.3±25.2 | 10.1±18.1 | 6.3±15.6 | 19.7±23.7 | 16.5±23.7 | 9.0±15.6 |
| | RCP8.5 | 6.8±19.4 | 5.3±18.8 | 1.1±15.1 | 10.5±22.8 | 17.4±21.9 | 9.8±14.7 |

4.    Are the scientific methods and assumptions valid and clearly outlined?

Yes, the methodology adopted in the paper is widely accepted and clearly described.

Response: Thanks for your positive feedback and comment.

5.    Are the results sufficient to support the interpretations and conclusions?

Yes, but uncertainty associated with projections should also be highlighted in the conclusions to provide a clearer and more comprehensive understanding to the readers. It is allowed readers or policymakers to interpret the results in a more informed manner.

Response: Thanks for your positive feedback and helpful comment. In the revised manuscript, we have quantified the uncertainty of future projections using the standard deviation of the runoff results from GHM (e.g., WaterGAP2) under four GCMs drives. Following the above comment, the uncertainty range has been marked in the text in the revised manuscript when quantifying future runoff projections:

Abstract:
Line 12 : "*Under representative concentration pathways (RCPs, i.e., RCP2.6, RCP6.0 and RCP8.5), runoff of the MR is projected to increase significantly (p>0.05), e.g., $3.81\pm3.47$ m$^3$ s$^{-1}$ a$^{-1}$ ( $9\pm8\%$ increase in 100 years) at the upstream station under RCP2.6 and $16.36\pm12.44$ m$^3$ s$^{-1}$ a$^{-1}$ ($13\pm10\%$ increase in 100 years) at the downstream station under RCP6.0.*"

Section 3.3 ISIMIP future projections
"*For example, under the RCP2.6 scenario (see the first column of Fig. 6), the annual runoff changing rate of the upstream N2 station, midstream N5 station, and downstream N8 station increased from $3.81\pm3.47$ m$^3$ s$^{-1}$ a$^{-1}$ to $7.40\pm7.41$ m$^3$ s$^{-1}$ a$^{-1}$ and final to $12.94\pm11.41$ m$^3$ s$^{-1}$ a$^{-1}$, respectively.*"

……

More about the places marked with uncertainty can be found in the revised manuscript.

6.    Is the description of experiments and calculations sufficiently complete and precise to allow their reproduction by fellow scientists (traceability of results)?

Yes. The methods and data used in the study are sufficiently explained, thus enabling other researchers to reproduce the results.

Response: Thanks for your positive feedback and comment.

7.    Do the authors give proper credit to related work and clearly indicate their own new/original contribution?

Yes, the paper includes a substantial number of cited references. However, the authors fail to highlight how their results or approach is different from some of the similar studies conducted in the past. While they have conducted a comprehensive analysis of historical and future runoff in the Mekong River Basin, it is essential to clearly articulate the novel contributions of their work in relation to the existing literature.

Response: Thanks for your helpful comments. We apologize for the unclear elaboration of the contribution/novelty of the manuscript. Similar to our response to Specific comments #2, we have appropriately emphasized the contributions of our work in the revised manuscript. Please see our responses to your Comment #2.

8.      Does the title clearly reflect the contents of the paper?

Yes.

Response: Thanks for your positive feedback and comment.

9.      Does the abstract provide a concise and complete summary?

Yes.

Response: Thanks for your positive feedback and comment.

10.      Is the overall presentation well-structured and clear?

Yes, the paper is well structured. However, the discussion section seems to redundantly summarize the results without delving into new insights. To enhance the paper's impact, the authors should utilize the discussion section to highlight how their results offer fresh perspectives and novel contributions concerning the future runoff of MRB beyond what has already been published. By focusing on these unique insights, the authors can provide a more substantial context for their findings.

Response: Thanks for your helpful comments and suggestions. We have made important revisions to the conclusion to emphasize our new findings and improve the conclusion section in the revised manuscript:

4 Discussion
Line 220 :" *4 Discussion*
  *Our results show a decreasing trend in the upstream and downstream runoff in the MRB and an increasing trend in the midstream runoff over the past 50 years. Of the 8 stations, only the N3 station reached a significant level of change. Hydropower energy development is one of the important human activities of MRB, which profoundly affects the runoff behavior of the basin. The 1990s are the early days of the establishment of the reservoirs, which significantly reduces the annual runoff. The subsequent operation*

*period of the reservoir has little impact on the interannual runoff, but mainly affects the distribution process of seasonal runoff.*

*In the MRB, the global climate models all perform well, except for the GFDL-ESM2M climate model. Moreover, GCM ensemble averaging can reduce the uncertainty of meteorological forcing. Meanwhile, based on the results of the GCM ensemble average, all GHMs have good runoff performance. Among these GHMs, WaterGAP2 performs the best, thanks to the calibration of the model (Chen et al., 2021). Overall, the runoff simulation results under the best combination are not inferior to the regional hydrological model. Under this combination, the R² of each station under the historical scenario (1971~2005) is about 0.75. Even in the historical simulation stage (2006~2020) under the RCPs scenario, it has the same performance. The satisfactory simulation performance of runoff enhances our confidence in future runoff analysis, and also provides a tool to understand the evolution law of future runoff.*

*This study systematically analyzes the performance and uncertainty of runoff simulations from five GHMs driven by four GCMs within the MRB during historical periods. An interesting finding is that the variability introduced by the GCMs was similar to or even greater than that introduced by the GHMs on the runoff simulation (Figure 3 and Figure 4). For example, in Figure 3a, the median $R^2$ of different GHMs under the same GCM driver can differ by 0.20, but the median $R^2$ of the same GHM under different GCMs drivers can differ by more than 0.20. To reduce the variability of runoff simulation, on the one hand, we can obtain a well-performing GHM through comprehensive evaluation. In this study, three performance indicators are combined under eight hydrological stations, and WaterGAP2 (i.e., GHM) is found to have the best performance (highest $R^2$ and NSE, and lowest Pbias) under four GCM drivers in the MRB. In addition, even a good GHM has high uncertainty for future runoff projection under different GCM drivers. A feasible approach at this time should be to combine the ensemble average of runoff results from the GHM driven by different GCMs, which can help reduce the uncertainty from climate models in future projections. At the same time, the standard deviation of runoff results from the GHM driven by different GCMs can be used to quantify the uncertainty in future runoff projections.*

*Another point is that under different RCPs, the interannual runoff of the three representative sites has a significant (p<0.05) increasing trend, which is consistent with the previous relevant studies suggesting that MRB runoff would increase in the future due to climate change (Hoang et al., 2019; Liu et al., 2022). A novel finding is that the upstream, midstream and downstream stations in the MRB show different change patterns under different RCP scenarios. Among these stations, under the same RCP, the runoff increasing rate of downstream stations will be higher than that of upstream station, which is consistent with the known understanding of the routing process. Interestingly, the runoff change behaviour of the same representative station under different RCPs is not consistent. The increase in runoff at upstream station N2 increased sequentially as the scenarios changed from RCP2.6 to RCP6.0 then to RCP8.5. The*

*difference is that the downstream station N8 has the highest runoff increase under the RCP6.0 scenario, while not under the RCP8.5 scenario. This behavior is closely related to the combined effects of temperature and precipitation on runoff under different RCP scenarios. Specifically, at upstream stations, the synergistic effect of increased glacial meltwater and increased precipitation caused by warming under different scenarios is greater than the effect of increased evapotranspiration caused by warming. This results in the highest runoff increase under RCP8.5. At downstream stations, the proportion of glacier meltwater to total water volume decreased, suggesting that its impact on total runoff was also lower. Moreover, as the downstream latitude decreases, the evapotranspiration increment caused by warming also increases. Under the combined effect of these factors, the runoff increases under the RCP6.0 scenario (16.36±12.44 m³/s/a) is higher than that under the RCP8.5 scenario (13.28±12.20 m³/s/a). This means that the risk of future flooding in the middle and lower reaches of the MRB is still likely to remain a high level, even if we try to manage to stay on a moderate emissions path (i.e., RCP6.0). The novel change patterns of the upper, middle and lower reaches explored in the study may be able to provide a scientific basis for the future implementation of local water resource management schemes in each reach of the MRB.*"

11.    Is the language fluent and precise?

Yes.

Response: Thanks for your comment.

12.    Are mathematical formulae, symbols, abbreviations, and units correctly defined and used?

Yes.

Response: Thanks for your comment.

13.    Should any parts of the paper (text, formulae, figures, tables) be clarified, reduced, combined, or eliminated?

Yes, as I mentioned in previous comments, the discussion section should be improved and elaborated.

Response: Thanks for your helpful comments. We have made important revisions to the

conclusion to emphasize our findings and improve the conclusion section in the revised manuscript. The detailed revision of the conclusion in the revised manuscript can be found in our response to Specific comments #10.

14.     Are the number and quality of references appropriate?

Yes.

Response: Thanks for your comment.

15.     Is the amount and quality of supplementary material appropriate?

N/A

Response: Thanks for your comment.

References:
Chen, H., Liu, J., Mao, G., Wang, Z., Zeng, Z., Chen, A., Wang, K., and Chen, D.: Intercomparison of ten ISI-MIP models in simulating discharges along the Lancang-Mekong River basin, Sci Total Environ, 765, 144494, 10.1016/j.scitotenv.2020.144494, 2021.
Hoang, L. P., Lauri, H., Kummu, M., Koponen, J., van Vliet, M. T. H., Supit, I., Leemans, R., Kabat, P., and Ludwig, F.: Mekong River flow and hydrological extremes under climate change, Hydrology and Earth System Sciences, 20, 3027-3041, 10.5194/hess-20-3027-2016, 2016.
Hoang, L. P., van Vliet, M. T. H., Kummu, M., Lauri, H., Koponen, J., Supit, I., Leemans, R., Kabat, P., and Ludwig, F.: The Mekong's future flows under multiple drivers: How climate change, hydropower developments and irrigation expansions drive hydrological changes, Science of The Total Environment, 649, 601-609, 10.1016/j.scitotenv.2018.08.160, 2019.
Kingston, D. G., Thompson, J. R., and Kite, G.: Uncertainty in climate change projections of discharge for the Mekong River Basin, Hydrology and Earth System Sciences, 15, 1459-1471, 10.5194/hess-15-1459-2011, 2011.
Lee, D., Lee, G., Kim, S., and Jung, S.: Future Runoff Analysis in the Mekong River Basin under a Climate Change Scenario Using Deep Learning, Water, 12, 10.3390/w12061556, 2020.
Liu, J., Chen, D., Mao, G., Irannezhad, M., and Pokhrel, Y.: Past and Future Changes in Climate and Water Resources in the Lancang–Mekong River Basin: Current

Understanding and Future Research Directions, Engineering, 13, 144-152, 10.1016/j.eng.2021.06.026, 2022.

Shrestha, B., Cochrane, T. A., Caruso, B. S., Arias, M. E., and Piman, T.: Uncertainty in flow and sediment projections due to future climate scenarios for the 3S Rivers in the Mekong Basin, Journal of Hydrology, 540, 1088-1104, 10.1016/j.jhydrol.2016.07.019, 2016.

Wang, S., Zhang, L., She, D., Wang, G., and Zhang, Q.: Future projections of flooding characteristics in the Lancang-Mekong River Basin under climate change, Journal of Hydrology, 602, 10.1016/j.jhydrol.2021.126778, 2021.

Yun, X., Tang, Q., Wang, J., Liu, X., Zhang, Y., Lu, H., Wang, Y., Zhang, L., and Chen, D.: Impacts of climate change and reservoir operation on streamflow and flood characteristics in the Lancang-Mekong River Basin, Journal of Hydrology, 590, 10.1016/j.jhydrol.2020.125472, 2020.

---

## Author Response (AR2)

**Reviewer 1**

Thank you for inviting me to re-review the paper: "Historical and Projected Future Runoff over the Mekong River Basin" by Wang et al.

I am impressed with the level of detail and responses to the last set of requests. I can also see that the replies to Reviewer 2 are thoughtful and comprehensive.

Response: We appreciate the reviewer's positive feedback and insightful suggestions, which are very helpful for us to improve the manuscript.

I think the paper is broadly ready to be published. However, the authors might like to consider improving (or expanding) just slightly more the important conclusions of the new paragraph (in Discussion) starting "This study systematically analyses the performance and uncertainty of runoff simulations from five GHMs driven by four GCMs….".

My interpretation of this paragraph is that it is slightly easier to constrain GHMs than GCMs, because GHMs can be forced offline with known climatological data and their performance analysed compared to river flow records. However, ESMs have substantial differences, both for the contemporary periods and indeed for their projections of future change. The authors might like to write a little more on how to, potentially, overcome inter-ESM uncertainty. The points to raise are:

(1) For the contemporary period, it is possible to weight ESMs by various statistics (such as the properties of rainfall projections at their gridboxes along the Mekong River). Once a weighting is made, this could also be used to weight future projections by different ESMs. However, the risk with this is that an ESM that performs poorly in the current climate might be very accurate at projecting key changes in near-surface meteorology.

(2) The other option always open to refining future projections is the Emergent Constraints (EC) method. To be useful in the context of Mekong River runoff, then their needs to be a quantity X across ESMs that links to a further quantity Y in the future (i.e., inter-ESM, there is a regression between X and Y). Then, if measurements are available of quantity X – which again might be a statistic of rainfall affecting runoff – then the regression can be used to refine important variable Y. There are many papers on ECs that might help – e.g., the review of Hall et al "Progressing emergent constraints on future climate change".

I am happy for the paper to now be accepted. However, the authors might like to include some of the above with a slightly expanded Discussion section. This will point readers to future analyses/research that may remove some of the uncertainty ranges identified

in the current Wang et al manuscript.

Response: Thank you very much for your helpful suggestions. As you noted, the aim of this paragraph is to illustrate that uncertainty between GCMs could introduce uncertainty into future runoff projections. To address this, we have calculated the mean and standard deviation of several GCM-driven GHM simulation results, allowing us to reduce and assess the uncertainty. This is a common practice for reducing uncertainty among models, known as "model democracy" (Taylor et al., 2012; Collins et al., 2013). Your constructive suggestions are thoughtful and valuable, and the approaches you mention can hopefully further reduce uncertainty (Knutti et al., 2017; Brient, 2019; Hall et al., 2019; Schlund et al., 2020; Yang et al., 2017). Although comparing the effectiveness of different approaches in reducing uncertainty is beyond the scope of our current manuscript, additional elaboration on these approaches may be helpful in subsequent analysis/research to further reduce the uncertainty identified in our current work. Following your recommendations, we have supplemented this paragraph with additional information on promising practices for reducing uncertainty in GCMs. The detailed additions and revisions we have made to the manuscript are shown in red font below:

"*This study systematically analyses the performance and uncertainty of runoff simulations from five GHMs driven by four GCM ...... At the same time, the standard deviation of runoff results from the GHM driven by different GCMs can be used to quantify the uncertainty in future runoff projections. This approach gives equal weight to each GCM, often referred to as "model democracy", and has been widely used in climate impact assessments (Taylor et al., 2012; Collins et al., 2013). Another approach that can potentially reduce uncertainty is a weighting scheme that considers the performance of the GCMs (Knutti et al., 2017; Yang et al., 2017). The GCMs are weighted by different statistics in the past or present, and the weighting coefficients are applied to the future projections of the GCMs. However, there is a risk that a GCM that performs poorly in the current climate may perform better when environmental conditions are beyond the contemporary range of change (Yang et al., 2017). It is worth mentioning that a novel and promising approach to constrain uncertainties is the emergent constraints (ECs). The EC approach consists of statistical (emergent) relationships between an observable quantity (X) in the past or present climate and a quantity (Y) related to the future climate across GCMs (Brient, 2019; Hall et al., 2019; Schlund et al., 2020). Combining emergent relationships with observations can potentially reduce uncertainty in future projections, and several published ECs have shown us positive effects (Schlund et al., 2020; Shiogama et al., 2022). We encourage further experimentation with various approaches, including those described above, to overcome the uncertainty among GCMs in the MRB.*"

**References**

Brient, F.: Reducing Uncertainties in Climate Projections with Emergent Constraints: Concepts, Examples and Prospects, Advances in Atmospheric Sciences, 37, 1-15, 10.1007/s00376-019-9140-8, 2019.

Collins, M., Knutti, R., Arblaster, J., Dufresne, J.-L., Fichefet, T., Friedlingstein, P., Gao, X., Gutowski, W. J., Johns, T., and Krinner, G.: Long-term climate change: projections, commitments and irreversibility, 2013.

Hall, A., Cox, P., Huntingford, C., and Klein, S.: Progressing emergent constraints on future climate change, Nature Climate Change, 9, 269-278, 10.1038/s41558-019-0436-6, 2019.

Knutti, R., Sedláček, J., Sanderson, B. M., Lorenz, R., Fischer, E. M., and Eyring, V.: A climate model projection weighting scheme accounting for performance and interdependence, Geophysical Research Letters, 44, 1909-1918, 10.1002/2016gl072012, 2017.

Schlund, M., Lauer, A., Gentine, P., Sherwood, S. C., and Eyring, V.: Emergent constraints on equilibrium climate sensitivity in CMIP5: do they hold for CMIP6?, Earth System Dynamics, 11, 1233-1258, 10.5194/esd-11-1233-2020, 2020.

Shiogama, H., Watanabe, M., Kim, H., and Hirota, N.: Emergent constraints on future precipitation changes, Nature, 602, 612-616, 10.1038/s41586-021-04310-8, 2022.

Taylor, K. E., Stouffer, R. J., and Meehl, G. A.: An Overview of CMIP5 and the Experiment Design, Bulletin of the American Meteorological Society, 93, 485-498, 10.1175/bams-d-11-00094.1, 2012.

Yang, H., Zhou, F., Piao, S., Huang, M., Chen, A., Ciais, P., Li, Y., Lian, X., Peng, S., and Zeng, Z.: Regional patterns of future runoff changes from Earth system models constrained by observation, Geophysical Research Letters, 44, 5540-5549, 10.1002/2017gl073454, 2017.